# Protocadherin 15 suppresses oligodendrocyte progenitor cell proliferation and promotes motility through distinct signalling pathways

Yilan Zhen[1], Carlie L. Cullen [1], Raphael Ricci [1], Benjamin S. Summers[1], Sakina Rehman[2], Zubair M. Ahmed [2], Antoinette Y. Foster[3], Ben Emery [3], Robert Gasperini[1,4] & Kaylene M. Young [1✉]

Oligodendrocyte progenitor cells (OPCs) express protocadherin 15 (Pcdh15), a member of the cadherin superfamily of transmembrane proteins. Little is known about the function of Pcdh15 in the central nervous system (CNS), however, Pcdh15 expression can predict glioma aggression and promote the separation of embryonic human OPCs immediately following a cell division. Herein, we show that Pcdh15 knockdown significantly increases extracellular signal-related kinase (ERK) phosphorylation and activation to enhance OPC proliferation in vitro. Furthermore, Pcdh15 knockdown elevates Cdc42-Arp2/3 signalling and impairs actin kinetics, reducing the frequency of lamellipodial extrusion and slowing filopodial withdrawal. Pcdh15 knockdown also reduces the number of processes supported by each OPC and new process generation. Our data indicate that Pcdh15 is a critical regulator of OPC proliferation and process motility, behaviours that characterise the function of these cells in the healthy CNS, and provide mechanistic insight into the role that Pcdh15 might play in glioma progression.

[1] Menzies Institute for Medical Research, University of Tasmania, Liverpool St, Hobart 7000, Australia. [2] Department of Otorhinolaryngology Head and Neck Surgery, School of Medicine, University of Maryland, Baltimore, MD 21201, USA. [3] Jungers Center for Neurosciences Research, Department of Neurology, Oregon Health & Science University, Portland, OR 97239, USA. [4] School of Medicine, University of Tasmania, Liverpool St, Hobart 7000, Australia. ✉email: kaylene.young@utas.edu.au

Oligodendrocyte progenitor cells (OPCs) express the mitogenic cell surface receptor, platelet-derived growth factor receptor α (PDGFRα)[1] and the NG2 proteoglycan[2], and comprise the largest proliferating cell population in the adult rat[3] or mouse[4] central nervous system (CNS). Throughout life, OPCs differentiate into mature, myelinating oligodendrocytes (OLs)[4–9]. Adult OPCs have a relatively homogeneous distribution throughout the grey and white matter regions of the mature CNS[3] but constantly extend and retract processes and extrude lamellipodia to detect cues and interact with other cells in the environment[10–13]. Each OPC retains its discrete 3-dimensional domain within the mouse brain through a process of contact-mediated repulsion[10]. The loss of an OPC from the network, through differentiation or cell death, initiates the homoeostatic replacement of that cell by activating the proliferation and migration of adjacent OPCs[10]. In this way, OPCs can balance the need for new OLs to support mature CNS function or aid regeneration, with the need to maintain OPC density and distribution, while preventing uncontrolled cell growth. How OPCs regulate their basal level of proliferation and motility, while recognizing neighbouring OPCs to maintain their territories remains unclear.

Microarray[14] and RNA sequencing[15–17] studies identified a number of genes that are highly expressed by OPCs and could regulate motility while allowing contact-mediated signalling between OPCs. These included members of the protocadherin (Pcdh) family of transmembrane calcium-dependent adhesion proteins, including *Pcdhβ*, *Pcdhγ*, *Pcdh10, Pcdh15* and *Pcdh20*[15]. Pcdh family members facilitate contact-mediated signalling to regulate cell functions including neurite distribution, synapse density and cell survival in vivo[18]. Within the Pcdh family, *Pcdh15* is more highly expressed by OPCs than other CNS cell types[15–17], however, its capacity to regulate OPC function remains unclear. The role of Pcdh15 is best understood in the peripheral nervous system, where it interacts with cadherin 23 to form the tip-complex[19], a structure critical to hair cell mechanotransduction in the inner ear. Genetic variants in *PCDH15* that prevent protein binding to cadherin 23 cause Usher syndrome type 1F (USH1F), a disease characterized by retinitis pigmentosa, congenital deafness, and vestibular areflexia[20,21]. As *cadherin* 23 is not highly expressed by cells in the CNS[15], it is likely that Pcdh15 signals by a distinct mechanism within the CNS.

Gliomas are heterogeneous cancers that often comprise a mixture of cells with gene expression profiles resembling neural stem cells, astrocytes and OPCs[22–24]. Gliomas with a low level of *PCDH15* expression are associated with poorer patient survival outcomes[25,26], and xenografted grade IV glioblastoma multiforme cells produce nodular rather than diffuse tumours when they express low levels of *PCDH15*[27]. Low PCDH15 expression is driven by the overexpression of miR-22-5p, generated from the long non-coding RNA, *MIR22HG*, and overexpressing *Pcdh15* in glioma cells can reduce xenograft growth and improve mouse survival[28]. These data suggest that Pcdh15 is likely to influence the proliferative and migratory properties of OPCs.

Herein we show that Pcdh15 is highly expressed by OPCs, where it suppresses ERK phosphorylation to negatively regulate OPC proliferation. Pcdh15 can also modulate actin polymerization to facilitate lamellipodial and filopodial motility. More specifically, Pcdh15 inhibits cdc42 signalling and the downstream Arp2/3 complex to reduce F-actin accumulation in OPC lamellipodia and enable the rapid extrusion and retraction of lamellipodia and the rapid contact-mediated retraction of filopodia. Finally, Pcdh15 promotes process generation by OPCs—a behaviour that is reliant on Cdc42 signalling.

## Results

### Pcdh15 is expressed by OPCs

To determine whether OPCs express *Pcdh15*, we first performed in situ hybridization, to detect *Pdgfrα* and *Pcdh15* mRNA within the E16.5 mouse spinal cord (Fig. 1a–c). *Pdgfrα*+ OPCs were present throughout the spinal cord (Fig. 1b) and discrete *Pcdh15*+ cells could be clearly discerned throughout the white and grey matter (Fig. 1c). As OPCs are the major cell type present in the E16.5 spinal cord white matter, it was interesting to note that the size and distribution of *Pcdh15*+ cells resembled that of *Pdgfrα*+ OPCs in the white matter (Fig. 1c). In addition, the number of *Pdgfrα*+ OPCs (127 ± 7 cells per section) and *Pcdh15*+ cells (121 ± 2 cells per section) was equivalent in the E16.5 spinal cord white matter (mean ± SEM for $n = 3$ mice; unpaired $t$-test, $p = 0.6$). The broader expression pattern for *Pcdh15* within the grey matter may suggest that other cell types, such as neurons, also express *Pcdh15*. This in situ hybridization pattern is consistent with previously published data examining gene expression in acutely purified postnatal mouse cortical cells[15], which indicate that *Pcdh15* is highly expressed in OPCs, and at a lower level in neurons and newly differentiated OLs (Fig. 1d).

To further investigate the expression of *Pcdh15* within cells of the OL lineage in vivo, we sequenced RNA from the Sun1-eGFP+ nuclei of OPCs (*Ng2-CreER$^{T2}$:: Sun1-eGFP*) or mature OLs (*Plp1-CreERT:: Sun1-eGFP*), isolated from the adult mouse brain. OPCs predominantly expressed an isoform of Pcdh15 that included exons coding for the transmembrane domain and the final exon, which corresponded to the CD3 cytoplasmic domain[29] with an MTKL C-termini (Fig. 1e). *Pcdh15* expression was strongly downregulated with OL differentiation, being significantly reduced in nuclear RNA isolated from mature (PLP+) OLs (Fig. 1e). These data are consistent with previous reports that *Pcdh15* is highly expressed by OPCs in the postnatal mouse[14,15,17] and embryonic human brain[16]. To determine whether Pcdh15 expression is retained by OPCs in vitro, we isolated OPCs from primary glial cultures derived from the cortex of P0-P3 *Pdgfrα-histGFP* mice. At 9 days in vitro (DIV), immunocytochemistry confirmed that 99.97% ± 0.05% of GFP+ PDGFRα+ OPCs expressed Pcdh15 (Fig. 1f–h; 2935 cells counted across $n = 3$ independent cultures).

### Pcdh15 suppresses OPC proliferation in vitro

To investigate the capacity for Pcdh15 to regulate OPC function, we adopted an shRNA gene knockdown approach and reduced Pcdh15 expression in OPC primary cultures. OPCs were transfected with control (scrambled sequence) shRNA or a commercial mixture of 3 × *Pcdh15* shRNAs. After 12 hours (h), the cells were harvested to generate protein lysates. A western blot, performed using an antibody that detects all isoforms of Pcdh15 (anti-pan Pcdh15), revealed a similar array of protein bands in lysates generated from the adult mouse brain (Fig. 2a) and cultured primary OPCs (Fig. 2b). Pcdh15 variants that range in size from <50 kDa to ~250 kDa have been described for cochlear and retinal cells[29–33]. In brain and OPC lysates, we identified the ~60 KDa and ~180 kDa variants that have been previously reported for the brain[30]. The 60 KDa variant was the major form detected in primary OPCs and expression of this variant was reduced by ~96% in OPCs treated with *Pcdh15* shRNA (Fig. 2c; mean ± SEM, $n = 4$ independent cultures quantified).

While immunocytochemistry is a less robust method of protein quantification, it requires fewer primary cells, so we next determined whether it could be used as a proxy to demonstrate successful Pcdh15 knockdown. Naïve (untransfected) OPC cultures and those treated with control or *Pcdh15* shRNAs were processed to detect the OPC marker PDGFRα and Pcdh15

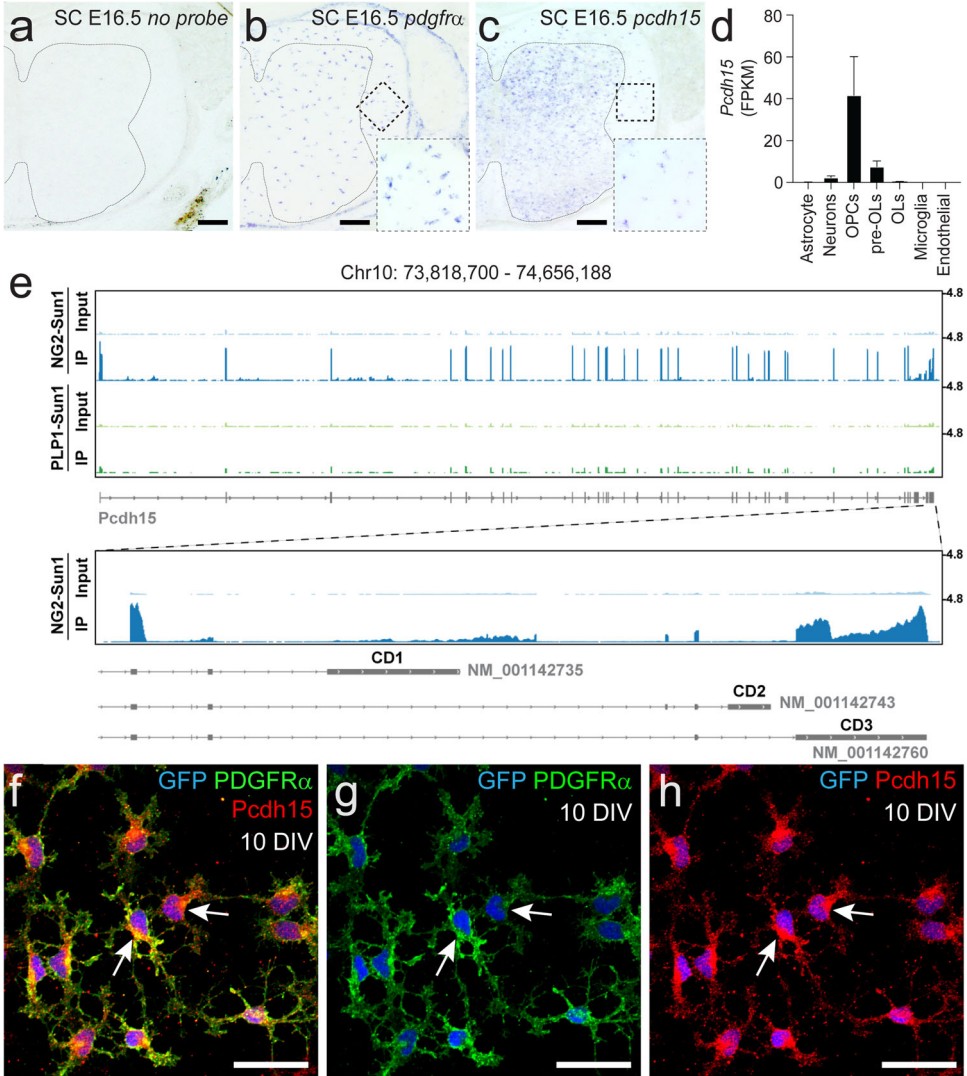

**Fig. 1 Pcdh15 mRNA and protein is expressed by OPCs. a–c** Transverse spinal cord sections from embryonic day 16.5 (E16.5) mice were used to perform in situ hybridization to detect *Pdgfrα* or *Pcdh15* mRNA. **a** E16.5 mouse spinal cord shows no labelling in the absence of an RNA probe. Line denotes the border of the spinal cord grey and white matter. **b** *Pdgfrα*⁺ OPCs are readily detected in the E16.5 spinal cord white matter. The dashed box denotes the white matter region enlarged in the lower right corner. **c** Image showing *Pcdh15*⁺ cells in the E16.5 mouse spinal cord. The dashed box denotes the white matter region enlarged in the lower right corner. **d** Quantification of *Pcdh15* mRNA (in FPKM, Fragments Per Kilobase Million) in neurons and glial cells acutely purified from the early postnatal mouse brain. Graph generated using previously published and publicly available RNA sequencing data from Zhang et al.[15]. **e** Genome browser tracks of the *Pcdh15* locus showing RNA-Seq alignments from nuclear RNA purified from the brains of adult *Ng2-CreER^T2:: Sun1-eGFP* or *PLP-CreERT:: Sun1-eGFP* mice. *Pcdh15* mRNA is highly expressed by *Ng2*⁺ OPCs and most transcripts correspond to the CD3 isoforms of Pcdh15. **f–h** Immunocytochemistry to detect GFP (blue), PDGFRα (green) and Pcdh15 (red) in primary OPCs generated from the early postnatal *Pdgfrα-histGFP* mouse cortex. Scale bars represent 60 μm (**a–c**) and 20 μm (**f–h**).

(Fig. 2d–f). By quantifying the integrated pixel density for Pcdh15 immunofluorescence, we determined that OPCs in the *Pcdh15* shRNA-treated cultures had ~36% less fluorescence than control cultures (Fig. 2g; mean ± SEM; $n = 3$ independent cultures analysed per group and ≥136 cells analysed per culture; one-way ANOVA, $p = 0.0005$). Furthermore, treating OPC cultures with control shRNA or one of the three components of the commercial *Pcdh15* shRNA in isolation (referred to as A-C), produced a similar reduction in Pcdh15 fluorescence (Fig. 2h–l; ~33%, ~32% and ~30% less fluorescence than control OPCs, respectively). This immunocytochemical approach clearly under-estimates the level of knockdown but was used to confirm knockdown in all subsequent functional experiments.

To evaluate the effect that Pcdh15 knockdown has on OPC proliferation in vitro, 10DIV control- and *Pcdh15*-shRNA treated

cultures were exposed to the thymidine analogue, EdU (10 μM), for 12 h before they were processed to detect GFP (green) and EdU (red) (Fig. 3a–f). We found that approximately twice as many GFP⁺ OPCs had divided and incorporated EdU in Pcdh15 knockdown than equivalently treated control cultures (Fig. 3g; mean ± SEM, $n = 3$ independent cultures and ≥603 OPCs quantified per coverslip; unpaired *t*-test, $p = 0.048$). A similar increase in OPC proliferation was observed when OPC cultures were treated with any of the 3 *Pcdh15* shRNAs (A-C) in isolation (Fig. 3h–p; mean ± SEM, $n = 3$ independent cultures and ≥791 OPCs quantified per coverslip). A more detailed analysis of the EdU⁺ GFP⁺ nuclei revealed that they more frequently appeared as doublets in Pcdh15 knockdown cultures, i.e., pairs of EdU⁺ GFP⁺ nuclei in direct contact (Fig. 3q), and the mean Euclidean distance between EdU⁺ GFP⁺ cells was reduced (Fig. 3r). These

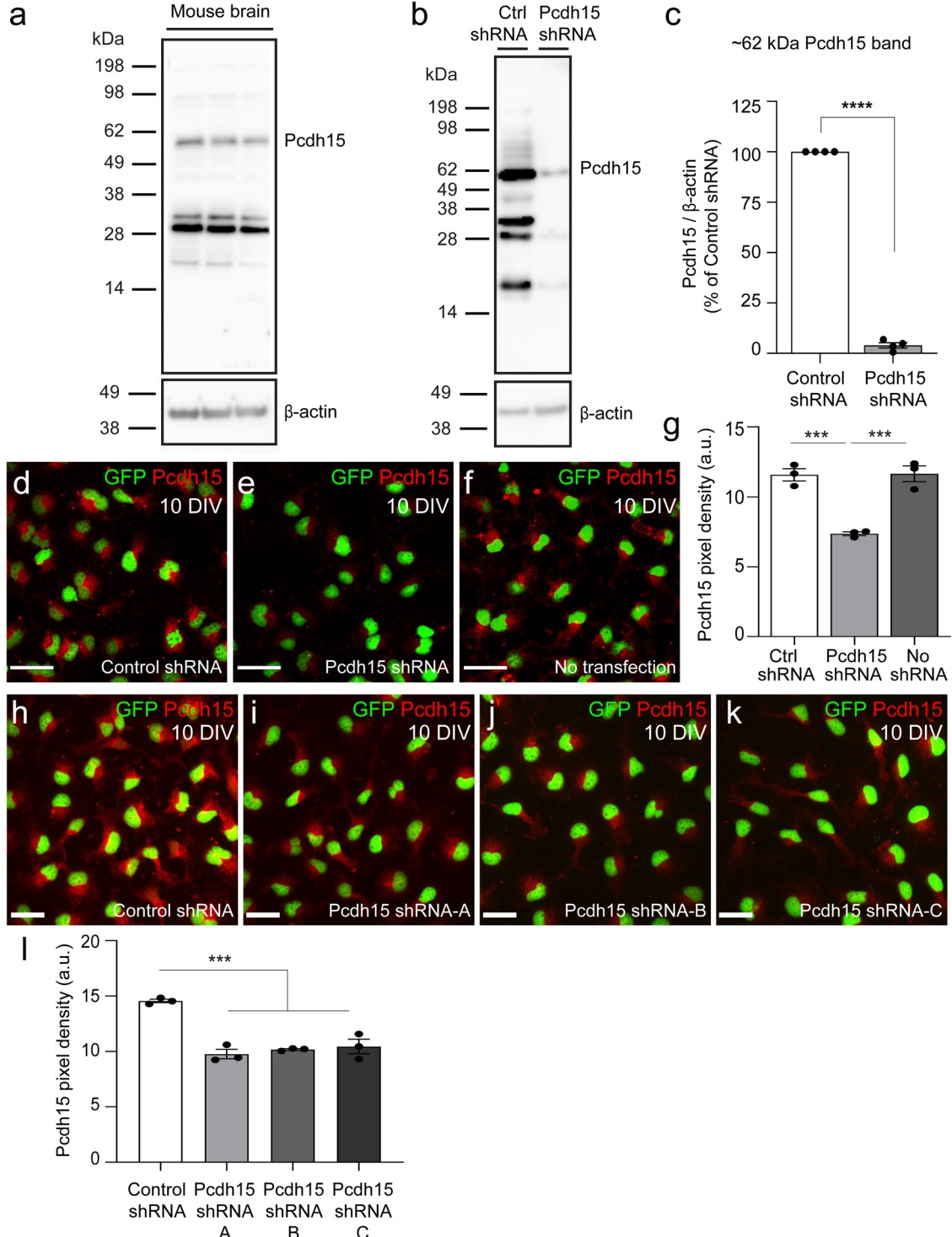

data are consistent with Pcdh15 acting as a negative regulator of OPC proliferation in vitro.

Several studies have shown that the canonical Wnt (wingless and integration site) signalling pathway, involving β-catenin[34,35], and the extracellular regulated kinase (ERK) signalling pathway[36,37] are intracellular regulators of OPC proliferation. To determine whether the canonical Wnt signalling pathway is regulated by Pcdh15 in OPCs, we collected cell lysates from OPCs transfected with control or *Pcdh15* shRNAs and performed a western blot to detect β-catenin and the phosphorylated (inactive) form of β-catenin. We determined that neither β-catenin or

phosphorylated β-catenin (serines 33, 37 or threonine 41; p-β-catenin) expression was significantly altered by Pcdh15 knock-down (Fig. 4a–c). These data suggest that β-catenin activity is not robustly regulated by Pcdh15. By performing a Western blot to detect ERK1 and 2, as well as the active, phosphorylated forms of ERK1 and ERK2 (p-ERK1 and p-ERK2) (Fig. 4d), we determined that Pcdh15 knockdown did not affect total ERK1 or ERK2 protein levels, relative to GAPDH expression [mean ERK1/GAPDH ± SEM for control-shRNA (1.61 ± 0.21) and *Pcdh15*-shRNA (1.76 ± 0.05) treated cultures, $n = 3$ independent cultures, unpaired $t$ test, $p = 0.55$; mean ERK2/GAPDH ± SEM for control

**Fig. 2 Pcdh15 shRNA treatment reduces Pcdh15 protein expression in cultured OPCs. a** Western blot gel image showing Pcdh15 expression in protein lysates generated from adult mouse brain ($n = 3$ mice). **b** Western blot gel image showing control- and Pcdh15-shRNA treated 9DIV OPC cultures, 12 h after transfection. **c** Expression (pixel integrated density) of the ~62 KDa Pcdh15 variant was quantified relative to the expression of β-actin in protein lysates from control and Pcdh15 knockdown OPCs, 12 h after transfection. Data are normalised to the relative expression of Pcdh15 in control OPCs for each culture [mean ± SEM, $n = 4$ independent cultures; one sample $t$-test for Pcdh15-shRNA: deviation from 100 (%) = −96.02, SD of deviation = 2.54, $t(3) = 75.33$, $p < 0.0001$]. **d–f** Confocal images of control- and *Pcdh15*-shRNA treated and untransfected OPCs following immunocytochemistry to detect Pcdh15 (red) and GFP (OPCs, green). **g** Quantification of Pcdh15 protein expression (immunocytochemistry pixel integrated density) in control- and Pcdh15-shRNA treated and untransfected OPCs [mean ± SEM, $n = 3$ independent cultures; one-way ANOVA: $F_{(2,6)} = 33.83$, $p = 0.0005$; Tukey's multiple comparisons test: ***$p = 0.001$, control shRNA vs. no transfection $p = 0.99$]. **h–k** Confocal images showing OPCs transfected with control shRNA or one of three distinct Pcdh15-shRNAs (Pcdh15-shRNA-A or Pcdh15-shRNA-B or Pcdh15-shRNA-C) for 12 h, before immunocytochemistry was performed to detect Pcdh15 (red) and GFP (OPCs, green). **l** Quantification of Pcdh15 expression (integrated pixel density) for OPCs transfected with control shRNA, Pcdh15-shRNA-A, Pcdh15-shRNA-B or Pcdh15-shRNA-C [mean ± SEM, $n = 3$ independent cultures; one-way ANOVA: $F_{(3,8)} = 31.05$, $p < 0.0001$; Tukey's multiple comparisons test: ***$p < 0.001$; Pcdh15-shRNA-A vs. B and B vs. C, $p = 0.9$; Pcdh15-shRNA-A vs. C $p = 0.6$]. Scale bars represent 30 μm (**d–f**) or 20 μm (**h–k**).

shRNA (1.73 ± 0.52) and *Pcdh15* shRNA (1.73 ± 0.33) treated cultures, $n = 3$ independent cultures, unpaired $t$ test, $p = 0.99$]. However, Pcdh15 knockdown was associated with a significant increase in the proportion of ERK1/2 that was phosphorylated (e.g. p-ERK1/total ERK1 and p-ERK2/total ERK2; Fig. 4e).

To determine whether the increased proliferation detected in *Pcdh15* knockdown OPC cultures was reliant on ERK1/2 phosphorylation, we treated control and Pcdh15 knockdown cultures with DMSO (diluent) or the highly selective MEK1/2 inhibitor, U0126[38,39], from the time of lentiviral transfection. At a concentration of 10 μM U0126 has been shown to prevent MEK1/2-dependent ERK phosphorylation in primary OPC cultures[40,41]. After 12 h of treatment, protein lysates were generated from the OPC cultures, and a western blot analysis revealed that U0126 treatment had no effect on total ERK1 or ERK2 expression, relative to GAPDH expression [mean ERK1/GAPDH ± SEM from $n = 3$ independent cultures: control-shRNA (0.54 ± 0.11) vs *Pcdh15*-shRNA (0.63 ± 0.08) vs control-shRNA + U0126 (0.62 ± 0.10) vs *Pcdh15*-shRNA + U0126 (0.60 ± 0.06); two-way ANOVA with Bonferroni multiple comparisons test for ERK1/GAPDH: shRNA $F_{(1,8)} = 0.13$, $p = 0.73$; drug treatment $F_{(1,8)} = 0.06$, $p = 0.82$; interaction $F_{(1,8)} = 0.33$, $p = 0.58$. Mean ERK2/GAPDH ± SEM from $n = 3$ independent cultures: control-shRNA (0.85 ± 0.18) vs *Pcdh15*-shRNA (0.92 ± 0.07) vs control-shRNA + U0126 (0.95 ± 0.11) vs *Pcdh15*-shRNA + U0126 (0.94 ± 0.06); two-way ANOVA with Bonferroni multiple comparisons test for ERK2/GAPDH: shRNA $F_{(1,8)} = 0.08$, $p = 0.79$; drug treatment $F_{(1,8)} = 0.30$, $p = 0.60$; interaction $F_{(1,8)} = 0.11$, $p = 0.72$]. However, U0126 treatment markedly reduced ERK1 and ERK2 phosphorylation in control and *Pcdh15* shRNA treated OPCs (Fig. 4f, g).

To determine whether blocking ERK1/2 phosphorylation could offset the proliferative effect of Pcdh15 knockdown, additional cultures treated with diluent or U0126 were exposed to EdU for 12 h post-transfection to quantify proliferation. GFP⁺ EdU⁺ cells were clearly visible in all culture conditions (Fig. 4h–k). However, U0126 treatment significantly reduced the proportion of OPCs that became EdU-labelled in control and *Pcdh15* shRNA-treated cultures and prevented Pcdh15 knockdown from increasing the proportion of OPCs that were EdU-labelled (Fig. 4l). Modulating the expression of mitogenic receptors, such as PDGFRα, is one way that Pcdh15 could suppress ERK1/2 phosphorylation and OPC proliferation, however, Pcdh15 knockdown did not produce a gross change in PDGFRα expression in OPCs (see Supplementary Fig. 1). These data suggest that Pcdh15 is a negative regulator of ERK phosphorylation and OPC proliferation in vitro.

**In OPCs, Pcdh15 promotes contact-mediated filopodial repulsion and lamellipodial extrusion.** In the healthy adult mouse brain, OPC proliferation and migration are restricted by a process of OPC self-repulsion that involves individual OPCs withdrawing when they contact an adjacent OPC[10]. OPC self-repulsion is also seen in vitro. In culture, OPCs are highly dynamic, extending and retracting entire processes, actin-rich lamellipodia-like "veils" and thin filopodia. When the filopodia of adjacent OPCs come into contact, the filopodia are rapidly retracted. To determine whether Pcdh15 regulates the contact dependent or contact independent motility of OPCs, we collected time-lapse movies, capturing the movement of control shRNA- and *Pcdh15* shRNA-treated OPCs over 2 h. By visualizing filopodia extended by OPCs and identifying those that contact adjacent cells (Fig. 5a), we were able to measure filopodial contact time (from first contact until withdrawal). We found that filopodial contact time was significantly increased in Pcdh15 knockdown relative to control cultures (Fig. 5b; $n = 3$ independent cultures and ≥66 filopodial contacts examined per treatment condition), suggesting that Pcdh15 signalling facilitates recognition and/or withdrawal. We also imaged individual OPC processes to visualise the extrusion and retraction of lamellipodia-like "veils" (Fig. 5c). A complete round of extrusion and retraction was measured as the "veiling time", which was significantly longer in Pcdh15 knockdown cultures (Fig. 5d; $n = 3$ independent cultures and ≥146 processes analysed per treatment condition), suggesting that Pcdh15 signalling speeds up OPC basal motility and surveillance behaviours.

To quantify the mean number of processes supported by control or Pcdh15 knockdown OPCs, we exported and analysed images of OPCs from a single timepoint (Fig. 5e). We found that control OPCs support ~5 processes, while Pcdh15 knockdown OPCs support ~3 processes each (Fig. 5f). By following individual OPCs over time and quantifying the number of times that each cell generates and extends a new process from the soma, we were able to determine that Pcdh15 knockdown OPCs elaborate fewer new processes per hour than control OPCs (Fig. 5g, h; $n = 3$ independent cultures and ≥35 OPCs analysed per treatment condition per culture). While these data suggest that Pcdh15 knockdown OPCs have an abnormal morphology and altered process kinetics, this phenotype may relate to a broader change in OPC basal motility. Under proliferative culture conditions, the soma of each OPC does not move a great distance. However, by mapping the migration trajectories of individual OPCs (following the movement of the cell soma over time; Fig. 5i) we determined that the speed of soma movement (Fig. 5j), the total distance each cell moved (Fig. 5k) and the Euclidean distance i.e. total distance from soma start to end position (Fig. 5l) were all reduced in Pcdh15 knockdown cultures.

ERK signalling can influence the motility of multiple cell types[42]. In particular, a PDGF-A driven increase in p-ERK can facilitate

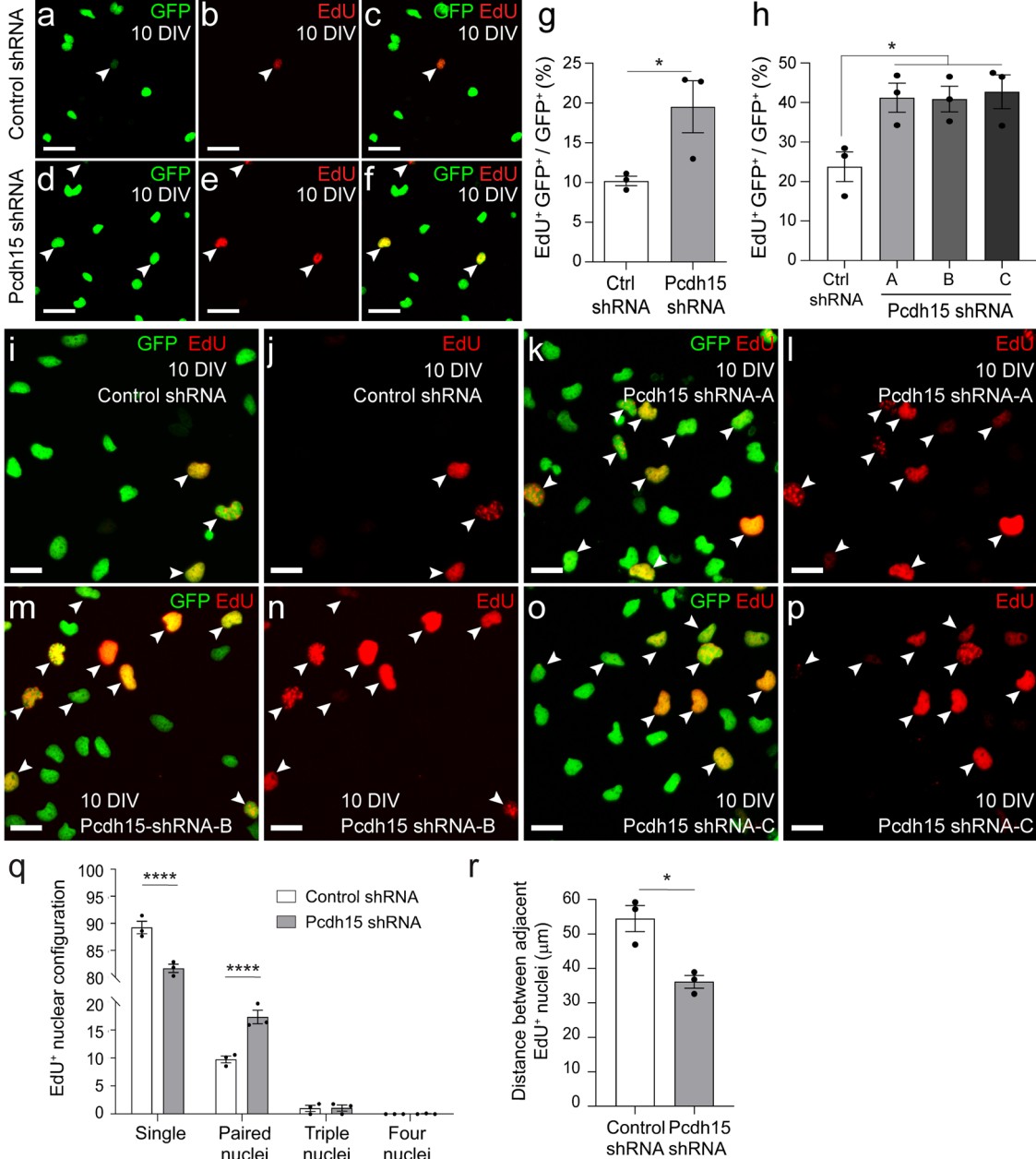

**Fig. 3 Pcdh15 knockdown increases OPC proliferation in vitro. a–f** Confocal images showing control (**a–c**) or Pcdh15 knockdown (**d–f**) GFP⁺ OPCs (green) after 12 h of EdU labelling (red). White arrowheads indicate EdU⁺ GFP⁺ OPCs. **g** The proportion of control and Pcdh15 knockdown GFP⁺ OPCs (%) that incorporated EdU (EdU⁺ GFP⁺/GFP⁺ × 100; mean ± SEM, $n = 3$ independent cultures; unpaired $t$-test, *$p = 0.048$). **h** The proportion (%) of GFP⁺ OPCs that incorporate EdU (EdU⁺ GFP⁺/GFP⁺ × 100) in cultures transfected with control-shRNA, Pcdh15-shRNA-A, Pcdh15-shRNA-B or Pcdh15-shRNA-C [mean ± SEM, $n = 3$ independent cultures; one-way ANOVA: $F_{(3,8)} = 5.658$, $p = 0.02$; Tukey's multiple comparisons test: Control shRNA vs Pch15 shRNA-A, *$p < 0.05$; Pcdh15-shRNA-A vs. Pcdh15-shRNA-B, $p = 1.0$; Pcdh15-shRNA-A vs. Pcdh15-shRNA-C, $p = 1.0$, Pcdh15-shRNA-B vs. Pcdh15-shRNA-C, $p = 0.99$]. **i–p** Confocal images showing EdU-labelling (red) in GFP⁺ OPCs (green) transfected with control-shRNA (**i, j**), Pcdh15-shRNA-A (**k, l**), Pcdh15-shRNA-B (**m, n**) or Pcdh15-shRNA-C (**o, p**). **q** The proportion (%) of EdU⁺ GFP⁺ nuclei that belonged to: single cells (nuclei not contacting others); paired cells (two nuclei in contact), or clustered cells (groups of three or four nuclei in contact with each other) [mean ± SEM for $n = 3$ independent cultures; ≥86 EdU⁺ GFP⁺ cells analysed per coverslip. Two-way ANOVA: shRNA $F_{(1, 16)} = 0.0037$, $p = 0.9524$; cluster $F_{(3, 16)} = 6206$, $p < 0.0001$; interaction $F_{(3, 16)} = 35.79$, $p < 0.0001$; with Bonferroni multiple comparison test: ****$p < 0.0001$. **r** Quantification of the mean Euclidean distance between EdU⁺ GFP⁺ nuclei (mean ± SEM, $n = 3$ independent cultures; ≥86 EdU⁺ GFP⁺ cells analysed per coverslip, unpaired $t$-test with welch's correction, *$p = 0.024$). Scale bars represent 25 μm (**a–f**) or 15 μm (**i–l**).

actin reorganization and promote OPC migration[43], and p-ERK has also been implicated in OPC filopodial regulation[41]. Therefore, we explored the possibility that elevated ERK phosphorylation underpinned the motility changes observed in Pcdh15 knockdown OPCs. We found that filopodial contact time was again increased in Pcdh15 knockdown cultures. However, preventing ERK1/2

phosphorylation, by treating control and Pcdh15 knockdown cultures with U0126, had no effect on filopodial contact time (Fig. 6a, b; $n = 3$ independent cultures and ≥44 filopodial contacts examined per treatment condition). Veiling time was again protracted in Pcdh15 knockdown cultures and this phenotype was not impacted by U0126 treatment (Fig. 6c, d; $n = 3$

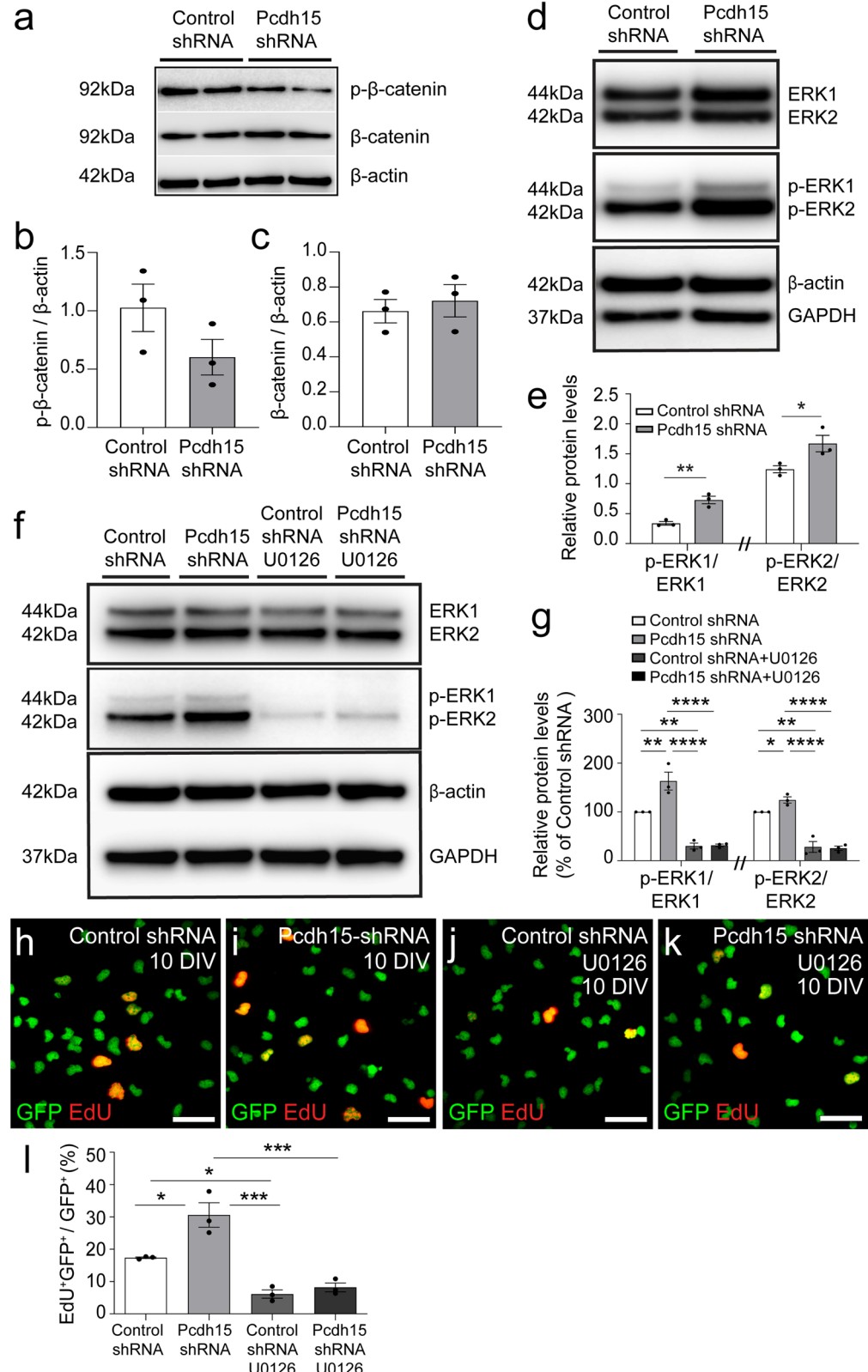

independent cultures per group and ≥102 processes analysed per treatment condition). Pcdh15 knockdown reduced the number of processes supported by individual OPCs from ~5 to ~3 processes per cell, but treatment with U0126 did not rescue this phenotype [example images in Fig. 6e; mean ± SEM for $n = 3$ independent cultures: Control-shRNA + diluent $(4.82 \pm 0.10)$ vs $Pcdh15$-shRNA + diluent $(3.016 \pm 0.18)$ vs Control-shRNA + U0126

$(4.72 \pm 0.05)$ vs $Pcdh15$-shRNA + U0126 $(3.022 \pm 0.09)$; two-way ANOVA: shRNA $F_{(1,8)} = 225.2$, $p < 0.0001$; drug $F_{(1,8)} = 0.17$, $p = 0.69$; interaction $F_{(1,8)} = 0.21$, $p = 0.65$; Bonferroni post-test: control-shRNA + diluent vs $Pcdh15$-shRNA + diluent, $p < 0.0001$; Control-shRNA + U0126 vs $Pcdh15$-shRNA + U0126, $p < 0.0001$; Control-shRNA diluent vs Control-shRNA + U0126, $p < 1.0$; $Pcdh15$-shRNA + diluent vs $Pcdh15$-shRNA + U0126, $p < 1.0$].

**Fig. 4 Pcdh15 suppresses OPC proliferation by reducing ERK phosphorylation. a** Primary mouse OPC cultures were transfected with control- or Pcdh15-shRNA and protein lysates collected 12 h later, at 10 DIV. Image of a Western blot performed to detect phosphorylated β-catenin (p-β-catenin), active unphosphorylated β-catenin, and β-actin. **b** Quantification of p-β-catenin (Ser33/Ser37/Thr41) expression by OPC cultures transfected with control- or Pcdh15-shRNA, normalised to β-actin expression (mean ± SEM, $n = 3$ independent cultures; unpaired $t$-test, $p = 0.17$). **c** Quantification of β-catenin expression by OPC cultures, 12 h after transfection with control- or Pcdh15-shRNA, normalised to β-actin expression (mean ± SEM, $n = 3$ independent cultures; unpaired $t$-test, $p = 0.63$). **d** Western blot gel image showing ERK1/2, phosphorylated ERK1/2, β-actin, and GAPDH expression by OPCs transfected with control- or Pcdh15-shRNA. **e** Western blot protein band integrated pixel density was quantified for p-ERK1/2 and ERK1/2 and used to calculate the proportion of ERK1 that was phosphorylated (p-ERK1/ERK1) and the proportion of ERK2 that was phosphorylated (p-ERK2/ERK2) for OPCs transfected with control- or Pcdh15-shRNA [mean ± SEM, $n = 3$ independent cultures; p-ERK1/ERK1 and p-ERK2/ERK2 data were analysed separately by unpaired $t$-test (note // break to $x$-axis), *$p < 0.05$, **$p < 0.01$]. **f** Western blot gel image showing ERK1/2, p-ERK1/2, β-actin, and GAPDH expression by OPCs transfected with control- or Pcdh15-shRNA and treated with DMSO or U0126 (10 μM). **g** Quantification of the proportion of ERK1 that was phosphorylated (integrated pixel density for p-ERK1/integrated pixel density for ERK1) and the proportion of ERK2 that was phosphorylated (integrated pixel density p-ERK2/integrated pixel density for ERK2) in OPCs transfected with control- or Pcdh15-shRNA and treated with DMSO or U0126. For each culture, these data were normalised to the control shRNA group, i.e., relative protein level/control shRNA relative protein level × 100 [mean ± SEM, $n = 3$ independent cultures; two-way ANOVA for p-ERK1/ERK1: shRNA treatment $F(1,8) = 11.06$, $p = 0.0104$; drug treatment $F(1,8) = 107.9$, $p < 0.0001$; interaction $F(1,8) = 10.18$, $p = 0.012$ with Bonferroni multiple comparisons, **$p < 0.01$, ****$p < 0.0001$; two-way ANOVA for p-ERK2/ERK2: shRNA treatment $F(1,8) = 2.12$, $p = 0.061$; drug treatment $F(1,8) = 91.19$, $p < 0.0001$; interaction $F(1,8) = 3.03$, $p = 0.032$ with Bonferroni multiple comparisons, *$p < 0.05$, ****$p < 0.0001$]. **h–k** Confocal images showing GFP⁺ OPCs (green) transfected with control- or Pcdh15-shRNA, treated with DMSO or U0126 and exposed to EdU (red) for 12 h. **l** The proportion (%) of GFP⁺ control or Pcdh15 knockdown OPCs that incorporate EdU following treatment with DMSO or U0126 [mean ± SEM, $n = 3$ independent cultures; two-way ANOVA: shRNA $F(1,8) = 13.0$, $p = 0.007$; drug treatment $F(1,8) = 62.98$, $p < 0.0001$; interaction $F(1,8) = 6.91$, $p = 0.03$ with Bonferroni multiple comparisons test: *$p < 0.05$, ***$p < 0.001$]. Scale bars represent 35 μm.

While the rate of new process generation by control shRNA OPCs exceeded that of Pcdh15 shRNA OPCs, this phenotype was unchanged by the addition of U0126 (Example images showing process generation in Supplementary Fig. 2, quantification in Fig. 6f; $n = 3$ independent cultures and ≥31 OPCs analysed per treatment condition per culture), indicating that process elaboration is not impacted by ERK1/2 phosphorylation.

**Pcdh15 knockdown promotes F-actin accumulation and elevates expression of the Arp2/3 complex.** OPC motility, including the structural changes involved in migration and differentiation, requires a dynamic regulation of the actin cytoskeleton[12,44–46]. To determine whether Pcdh15 regulates F-actin assembly in OPC processes, we performed immunocytochemistry to detect F-actin (phalloidin, red) and microtubules (α-tubulin, green) in control and *Pcdh15* shRNA-treated cultures (Fig. 7a). The level of F-actin present within each elaborated OPC veil was significantly increased in *Pcdh15* knockdown cultures compared to controls (Fig. 7a, b; $n = 4$ independent cultures and ≥51 veils analysed per treatment). However, Pcdh15 knockdown did not result in actin being polymerised within inappropriate regions of the OPC veil, as the level of colocalization between F-actin and α-tubulin was equivalent in control and *Pcdh15* shRNA treated cultures (Fig. 7c; $n = 3$ independent cultures and ≥37 veils analysed per treatment condition).

As the Akt signalling pathway regulates actin polymerisation in human glioblastoma multiforme cells and human glioma-initiating cells[47] and the Akt/mTOR (mechanistic target of rapamycin) pathway is a critical regulator of OPC differentiation, including actin polymerisation and depolymerisation[37,48], we explored the possibility that Pcdh15 influenced F-actin levels in OPC lamellipodia by regulating Akt expression or activity. However, we found that control and *Pcdh15* shRNA-treated OPCs expressed an equivalent level of Akt (Fig. 7d, e) and phosphorylated Akt (p-Akt; Fig. 7d, f), relative to β-actin expression.

To identify cytoskeletal regulators that were activated or inhibited by Pcdh15 signalling in OPCs, we performed a PCR array, comparing the expression of known cytoskeletal regulatory genes in control and Pcdh15 knockdown cultures. Twenty of the 84 cytoskeletal genes examined were found to be up or downregulated in Pcdh15 knockdown cultures (Fig. 7g; $n = 3$

independent cultures). Of these, *cyclin-dependent kinase 5 regulatory subunit 1* (*cdk5r1*; produces p35 and p25 proteins) and *cyclin-dependent kinase 5* (*cdk5*) are known regulators of circadian clock proteins and neurite outgrowth[49–51]. The serine/threonine protein kinase, MRCKα, encoded by upregulated gene *cdc42 binding protein kinase alpha* (*Cdc42bpa*), has also been implicated in lamellipodial actin dynamics, promoting LIM kinase 1 activity to increase phosphorylation of the actin severing protein, Cofilin (at serine 3) and consequently reducing F-actin depolymerisation[52–56]. Upregulated gene *Slingshot homologue 1* (*Ssh1*) encodes the opposing cofilin phosphatase, which dephosphorylates cofilin and promotes F-actin depolymerisation[56]. Within the lamellipodia, proteins encoded by *Nck adaptor protein 2* (*Nck2*), *C10 regulator of kinase* (*Crk*), *Rac Family Small GTPase 1* (*Rac1*), and *cell division control protein 42 homologue* (*Cdc42*) increase the activity of Wiskott Aldrich Syndrome protein (WASP) family members, which in turn activate the heptameric actin nucleator complex, Arp2/3, that includes proteins encoded by upregulated genes *actin-related protein* (*Actr*) *2, Actr3, Arpc* (*Actin Related Protein 2/3 Complex Subunit*) *2, Arpc3* and *Arpc5*[57–59].

As many of the detected changes in gene expression were small, we used the PCR array data as a guide, and performed a series of Western blots to quantify the expression of proteins involved in activating or forming the Arp2/3 complex in control and Pcdh15 knockdown OPC cultures (Fig. 7h–m). CrkII, an adaptor protein that complexes with p130-CAS and other proteins to activate WASP family members and promote Arp2/3 directed actin branching[60,61] was significantly elevated in Pcdh15 knockdown OPCs (Fig. 7h, i), as was WASP family member, WAVE2[62] (Fig. 7h, j). Consistent with the qPCR findings, the expression of Arp2 (Fig. 7h, k), Arp3 (Fig. 7h, l) and Arpc3 (Fig. 7h, m), components of the Arp2/3 signalling complex, were also elevated in Pcdh15 knockdown OPCs, suggesting that Pcdh15 is a negative regulator of Arp2/3-mediated actin nucleation.

As actin dynamics can be regulated by local protein activity, we next confirmed that Pcdh15 and members of the Arp2/3 complex were expressed throughout OPC processes. We performed immunocytochemistry to detect Pcdh15 (anti-pan Pcdh15, green) and Arp2 or Arp3 (red) in primary OPC cultures and collected high magnification, single-plane confocal

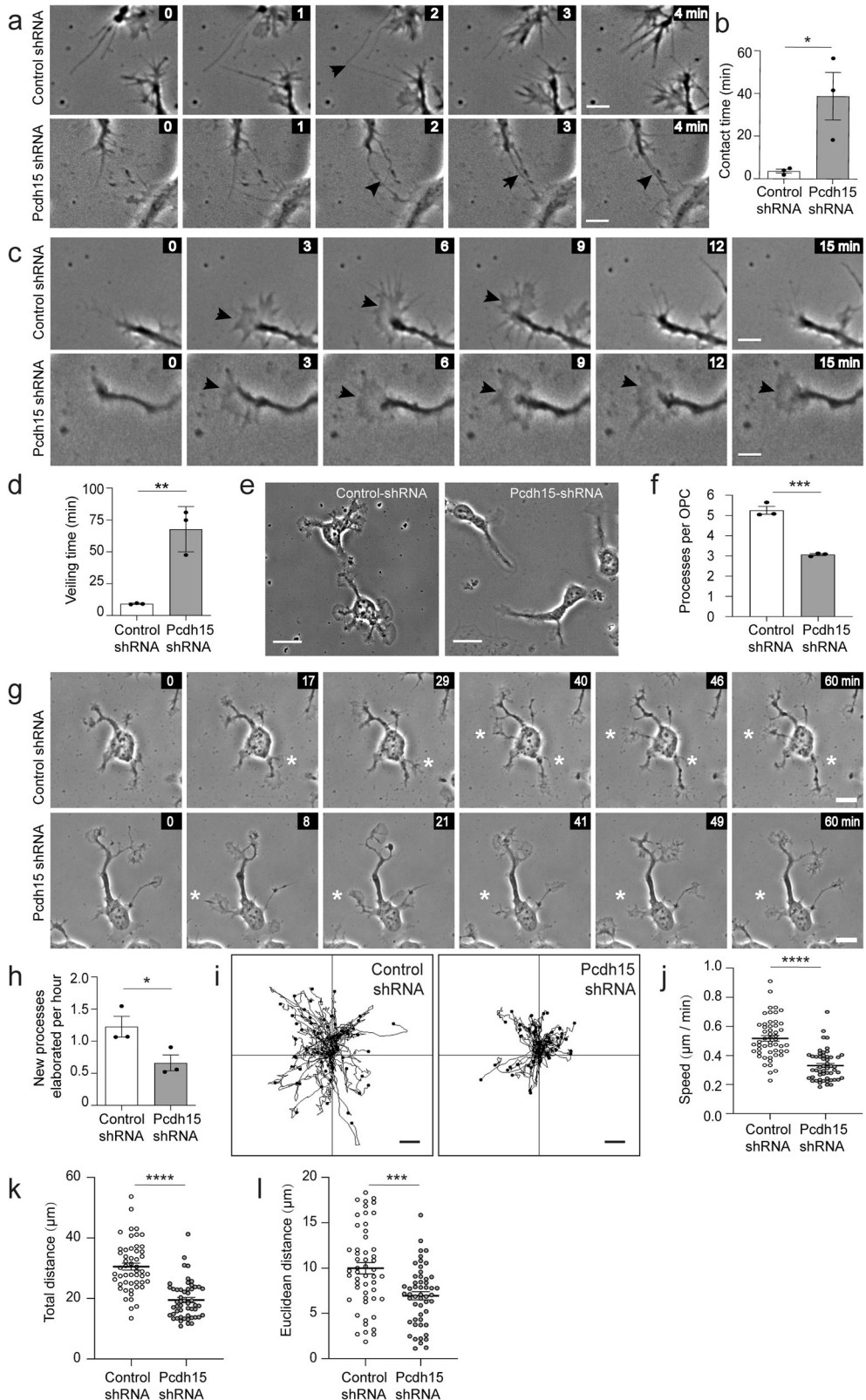

images for analysis (Fig. 8a–h). We found that Pcdh15 expression was highly concentrated on one side of the cell soma (Fig. 8a–c, e–g), but was also identified as immunofluorescent puncta along the length of each process (Fig. 8d, h). Puncta corresponding to Arp2 (Fig. 8a–d) and Arp3 (Fig. 8e–h) were also clearly visible in the OPC soma and along each process. This pattern of labelling was further confirmed by performing immunocytochemistry using antibodies to detect Arp2 or Arp3 and an antibody specific to CD3 isoforms of Pcdh15[29] (Fig. 8i–p), the major Pcdh15 isoforms expressed by OPCs (Fig. 1e). As Pcdh15 and Arp2/3 complex proteins are expressed at the OPC soma and within OPC processes, it is feasible that they are distinct components of a local signalling pathway that regulates actin dynamics in OPCs.

**Fig. 5 Pcdh15 promotes OPC motility and filopodial repulsion in vitro.** 12 h after OPCs were transfected with control or *Pcdh15*-shRNA, they were imaged once a minute for 2 h and the resulting time-lapse videos were used to quantify different aspects of OPC motility. **a** Image series show a filopodial contact (black arrowhead) between adjacent OPCs in a control and Pcdh15 knockdown culture. All images are time-stamped relative to the first image in the depicted series (designated as 0 min). **b** The mean duration (min) that a filopodial contact was sustained in control and Pcdh15 knockdown cultures (mean ± SEM, $n = 3$ independent cultures, $\geq 66$ filopodial contacts measured per condition; unpaired $t$ test, *$p = 0.034$). **c** Excepts from a time-lapse image series showing a control OPC process as it extrudes and retracts a veil and a Pcdh15 knockdown OPC as it extrudes and maintains its veil. The images are time-stamped relative to the first image in the depicted series (designated as 0 min). Black arrowheads indicate elaborated veils. **d** Quantification of the mean veiling time for OPC processes in control- and *Pcdh15*-shRNA treated cultures (time taken to complete an extrusion and retraction) (mean ± SEM, $n = 3$ independent cultures, $\geq 146$ veils analysed per condition; unpaired $t$-test, **$p = 0.004$). **e** Images of control- and *Pcdh15*-shRNA treated OPCs, 12 h after transfection. **f** Quantification of the mean number of processes supported by control and Pcdh15 shRNA-treated OPCs (mean ± SEM, $n = 3$ independent cultures and $\geq 81$ OPCs analysed per condition; unpaired $t$ test, ***$p = 0.0004$). **g** Short image series extracted from the time-lapse videos to show control and Pcdh15 knockdown OPCs extending a new process. All images are time-stamped relative to the first image in the depicted series (designated as 0 min) and the white asterisks denote the new processes. **h** Quantification of the mean number of new processes elaborated by control- and *Pcdh15*-shRNA transfected OPCs each hour (mean ± SEM, $n = 3$ independent cultures and $\geq 35$ OPCs analysed per condition; unpaired $t$ test, *$p = 0.048$). **i** The migration trajectories for the soma of 52 control- or *Pcdh15*-shRNA treated OPCs. **j** Mean migration speed for the soma of control- and *Pcdh15*-shRNA treated OPCs (mean ± SEM, $n = 52$ OPCs per group, unpaired $t$ tests with Welch's correction, ****$p < 0.0001$). **k** Mean distance that each OPC soma moved (path length) in control- and *Pcdh15*-shRNA treated cultures (mean ± SEM, $n = 52$ OPCs per group, unpaired $t$ tests with Welch's correction, ****$p < 0.0001$). **l** Mean Euclidean distance (shortest distance between the start and end points) that each OPC soma moved in control- or *Pcdh15*-shRNA treated cultures (mean ± SEM, unpaired $t$ tests with Welch's correction, ***$p < 0.0002$). Scale bars represent 5 µm (**a**, **c**), 8 µm (**g**) or 20 µm (**e**).

**Pcdh15 reduces Arp2/3 complex activity to facilitate contact-mediated filopodial repulsion and veiling kinetics.** To determine whether elevated Arp2/3 signalling is responsible for F-actin accumulation in Pcdh15 knockdown lamellipodia, we treated control and *Pcdh15* knockdown cultures with an Arp2/3 inhibitor, CK666 [50 µM[63]] or an inhibitor of the upstream Cdc42 GTPase, ML141 [10 µM[64]] at the time of lentiviral transfection (Fig. 9a). At these concentrations, CK666 and ML141 reduce F-actin nucleation in differentiating OPC and neuron cultures, respectively[44,65]. After 12 h, the cultures were processed to detect F-actin (phalloidin) and microtubules (α-tubulin) (Fig. 9b–m). Phalloidin expression was elevated in the veils of Pcdh15 knockdown OPCs, however, this phenotype was abrogated by treatment with CK666 or ML141 (Fig. 9n; $n = 3$ independent cultures and $\geq 50$ processes analysed per treatment condition). As both pharmacological inhibitors similarly reduced phalloidin expression in Pcdh15 knockdown cultures, further experiments only utilised ML141.

To determine whether Cdc42 inhibition can rescue the motility impairments detected in Pcdh15 knockdown OPCs, we collected time-lapse images of control and *Pcdh15* shRNA-treated OPCs cultured in the presence of diluent or ML141 (example image series in Fig. 9o). We found that the mean filopodial contact time was ~13 min between control OPCs but increased to ~32 min between Pcdh15 knockdown OPCs (Fig. 9p; $n = 3$ independent cultures and $\geq 61$ filopodial contacts analysed per treatment condition per experiment). However, treatment with ML141 reduced the mean filopodial contact time for Pcdh15 knockdown OPCs to ~8 min, a value that was not significantly different from control OPCs (Fig. 9p). Treatment with ML141 also restored OPC veiling time. By reviewing timelapse videos of OPC processes over time, as they extrude and retract lamellipodia-like veils, we were able to quantify the time taken for individual veils to be initiated, extruded and fully retracted (Fig. 9q, $n = 3$ independent cultures and $\geq 122$ processes measured per treatment condition). We found that the mean veiling time for vehicle-treated control OPCs was ~16 min and this increased to ~39 min in *Pcdh15* knockdown cultures (Fig. 9q; image series showing example veil extrusions and retractions are in Fig. 9r). When *Pcdh15* knockdown cultures were treated with ML141, veiling time was significantly shortened (~12 min; Fig. 9q), indicating that Pcdh15 is a negative regulator of

cdc42-Arp2/3 signalling and is necessary for the rapid motility of actin-rich OPC structures.

Pcdh15 knockdown also induced Arp2/3 independent changes in OPC motility. By performing timelapse imaging of control and Pcdh15 knockdown OPCs treated with diluent (DMSO) or ML141, we were able to analyse OPC morphology at a single time point or quantify the rate of new process generation (example image series showing new process generation in Fig. 10a). Control OPCs supported ~5 processes, while Pcdh15 knockdown OPCs supported ~3 processes, and this Pcdh15 knockdown phenotype was not rescued by ML141 treatment (Fig. 10b; $n = 3$ independent cultures and $\geq 34$ OPCs analysed per treatment condition per culture). Indeed, the application of ML141 reduced the number of processes supported by control OPCs (Fig. 10b), phenocopying the effect of Pcdh15 knockdown. Consistent with this effect, exposure to ML141 did not alter the number of new processes elaborated by Pcdh15 knockdown OPCs (Fig. 10a, c), but reduced new process generation by control OPCs, again phenocopying Pcdh15 knockdown (Fig. 10a, c). These data suggest that Cdc42 signalling supports process generation by OPCs and indicates that Pcdh15 signalling affects Cdc42 signalling differently at the soma and actin-rich lamellipodia.

## Discussion

Microarray[14] and RNA sequencing[15–17] data support a role for the transmembrane receptor, Pcdh15, in regulating OPC function, however, the it was unclear how Pcdh15 exerted this effect. Our research reveals that Pcdh15, particularly the CD3 isoform of Pcdh15, is highly expressed by OPCs (Figs. 1, 2 and 8), where it suppresses ERK1/2 phosphorylation and signalling, acting as a negative regulator of OPC proliferation (Figs. 3 and 4). Pcdh15 signalling also increases the mean number of processes supported by OPCs and enhances their basal motility, promoting the extension of new processes, the extrusion and retraction of lamellipodia-like veils from each process, and the rapid retraction of filopodia upon contact with an adjacent OPC (Fig. 5). Each behaviour is characteristic of OPC motility "at rest" as the cells sense and respond to their surrounding environment[10,13] and the associated morphological changes require modulation of the OPC cytoskeleton[66]. While ERK and Akt signalling have, in some contexts, been associated with OPC motility, preventing ERK phosphorylation had no impact on the motility of Pcdh15 knockdown OPCs, and Akt expression was not altered by Pcdh15

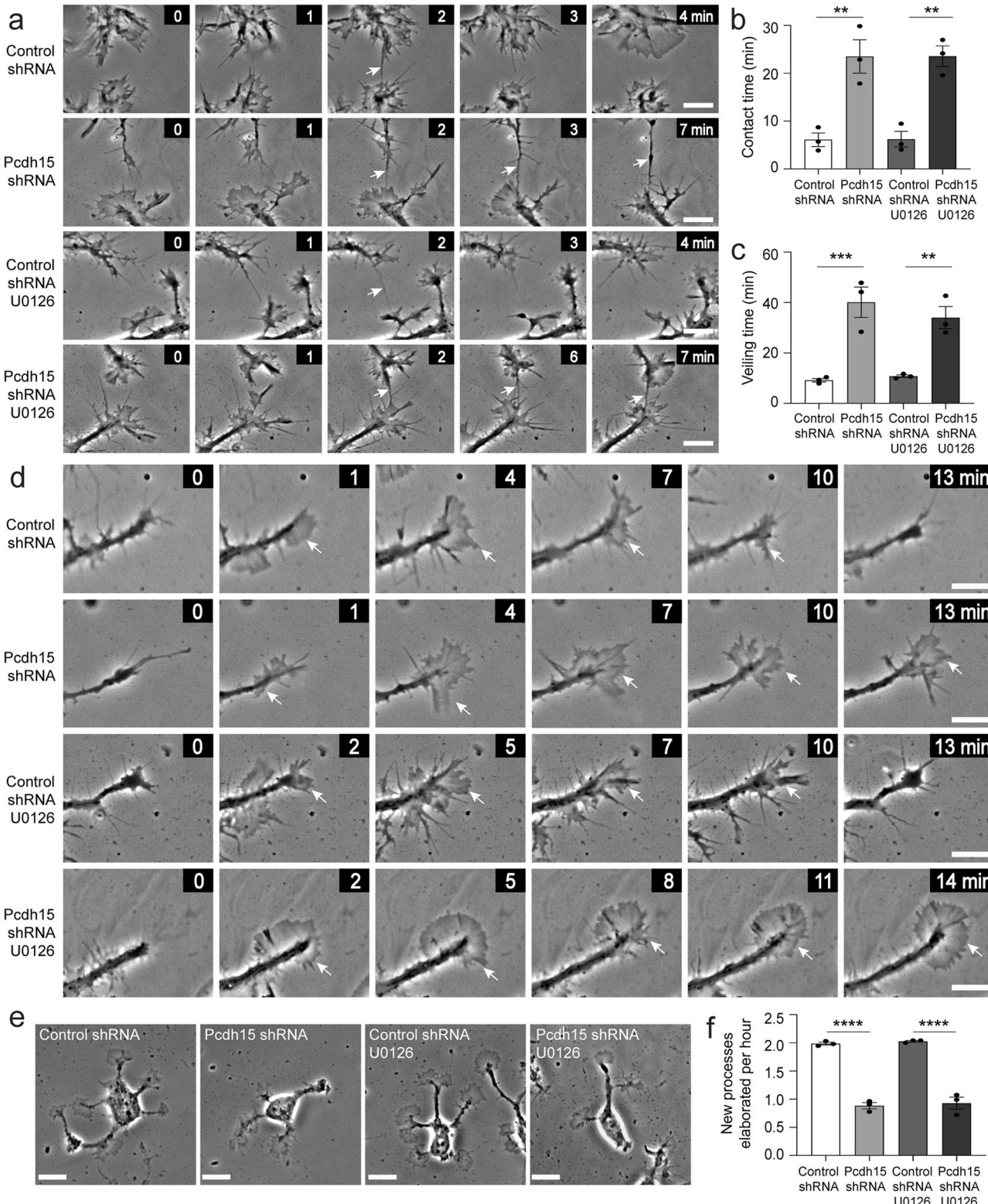

knockdown (Figs. 6 and 7). Instead, our qPCR and protein expression data indicated that Pcdh15 knockdown dysregulated ADF/Cofilin and Arp2/3 activity in OPCs (Fig. 7). Inhibiting Cdc42 GTPase activity upstream of Arp2/3 activity, or Arp2/3 activity directly, rescued actin accumulation in the lamellipodia to the same extent, and inhibiting Cdc42 rescued lamellipodial and filopodial dynamics in Pcdh15 knockdown cultures (Fig. 9). Of note, Cdc42 inhibition not only failed to return OPC process

number and the rate of new process generation to baseline levels in Pcdh15 knockdown cultures but reduced OPC process number and generation in control shRNA treated cultures, effectively mimicking the Pcdh15 knockdown phenotype (Fig. 10). These data suggest that Pcdh15 regulates different cytoskeletal signalling pathways at the soma (the site of process initiation) and within the OPC processes, where it locally suppresses Cdc42-Arp2/3 activity to promote veil extrusion and retraction.

**Fig. 6 Pcdh15-mediated inhibition of ERK does not influence filopodial repulsion or veiling kinetics. a** Image series extracted from time-lapse videos showing filopodial contacts (arrows) formed between adjacent OPCs in control- and *Pcdh15*-shRNA cultures treated with DMSO or U0126 (10 μM). All images are time-stamped relative to the first image in the depicted series (designated as 0 min). **b** The mean duration (min) that filopodial contacts were sustained once they formed between adjacent OPCs in control- and *Pcdh15*-shRNA cultures treated with DMSO or U0126 [mean ± SEM, $n = 3$ independent cultures, ≥44 filopodial contacts measured per condition; two-way ANOVA: shRNA treatment $F_{(1,8)} = 56.29$, $p < 0.0001$; drug treatment $F_{(1,8)} = 0.0015$, $p = 0.96$; interaction $F_{(1,8)} = 0.00017$, $p = 0.98$ with Bonferroni multiple comparisons test: **$p < 0.01$]. **c** Mean OPC process veiling time (min), i.e., the time taken for an OPC process to complete one full extrusion and retraction of its veil in control- or *Pcdh15*-shRNA cultures treated with DMSO or U0126 [mean ± SEM, $n = 3$ independent cultures, ≥102 veils analysed per condition; two-way ANOVA: shRNA treatment $F_{(1,8)} = 52.44$, $p < 0.0001$; drug treatment $F_{(1,8)} = 0.36$, $p = 0.56$; interaction $F_{(1,8)} = 1.05$, $p = 0.33$ with Bonferroni multiple comparisons, **$p < 0.01$, ***$p < 0.001$]. **d** Image series extracted from time-lapse videos showing OPC processes elaborating and retracting veils in control- and *Pcdh15*-shRNA cultures treated with DMSO or U0126. White arrows indicate extruded veils. All images are time-stamped relative to the first image in the depicted series (designated as 0 min) and images are selected to highlight veil extrusion and retraction. **e** Images of OPCs 12 h after transfection with control- or Pcdh15-shRNA and treatment with DMSO or U0126 (10 μM). **f** Quantification of the number of new processes elaborated by OPCs each hour in cultures transfected with control- or Pcdh15-shRNA and treated with DMSO or U0126 [mean ± SEM, $n = 3$ independent cultures, ≥31 OPCs analysed per condition; two-way ANOVA: shRNA treatment $F_{(1,8)} = 314.7$, $p < 0.0001$; drug treatment $F_{(1,8)} = 0.48$, $p = 0.50$; interaction $F_{(1,8)} = 0.0007$, $p = 0.97$ with Bonferroni multiple comparisons, ****$p < 0.0001$]. Image series depicting OPC process generation can be found in Supplementary Fig. 2. Scale bars represent 5 μm (**a**, **d**) or 10 μm (**e**).

As Pcdh15 knockdown was associated with a larger proportion of OPCs entering S-phase of the cell cycle to incorporate EdU over a 12 h labelling period (Fig. 3), our data indicate that Pcdh15 normally acts to suppress OPC proliferation in vitro. This is consistent with the role of other non-clustered PCDH family members, many of which have been identified as candidate tumour suppressor genes, including PCDH8[67,68], PCDH9[69], PCDH17[70], PCDH18[71], PCDH20[72] and, in some but not all cases, PCDH10[73–78]. Indeed, the overexpression of PCDH15 in glioma cell lines is sufficient to reduce proliferation and tumour growth following xenograft transplantation[28]. However, the capacity for Pcdh15 to regulate OPC proliferation may change between embryonic and postnatal development. In embryonic development, OPCs migrate as streams of dividing cells[79] and while OPC daughter cells are rapidly repelled from each other following division, there is no overt repulsion following process contact[16]. By contrast, in the early postnatal and adult CNS, OPC distribution is more homogeneous[3,4] and cell body movement is more restricted, but OPCs continually extend and retract processes—actively withdrawing a process when it contacts an adjacent OPC[10]. When Pcdh15 was knocked down in human embryonic OPCs, by treating the cells in vitro with shRNA or an anti-Pcdh15 antibody before transferring them into an embryonic organotypic cortical culture, newly generated daughter OPCs were less able to repel each other, so they remained in close proximity, and OPC proliferation was slightly reduced[16]. By contrast, when Pcdh15 was knocked down in postnatal OPC cultures in this study, the filopodia of adjacent OPCs were less able to retract following contact and OPC proliferation was significantly elevated (Figs. 3 and 5). This distinction may reflect differences in OPC maturity or the in vitro culture systems utilised.

OPC proliferation is enhanced by growth factors that activate the ERK1/2 signalling pathway, including FGF-2, PDGF-AA, IGF-1, BDNF and NT-3[36,80–83], and blocking ERK1/2 activity in OPCs, using pharmacological inhibitors, can prevent the mitogenic effect of these growth factors[36,81,84]. Furthermore, the conditional deletion of *Erk2* from *Olig2* expressing cells on an *Erk1* null background significantly decreases the number of OPCs detected in the E14.5 mouse spinal cord white matter and the proportion of OPCs that proliferate[85]. We found that *Pcdh15* knockdown resulted in a robust and rapid increase in ERK1/2 phosphorylation and OPC proliferation (Fig. 4 and Supplementary Fig. 3), which suggests that Pcdh15 suppresses the MAPK signalling pathway, either directly or by modulating the expression or activity of mitogenic receptors that activate the MAPK

signalling pathway. As Pcdh15 does not regulate Akt phosphorylation (Fig. 7), we predict that Pcdh15 would be less effective at suppressing OPC proliferation driven by growth factors that simultaneously promote ERK1/2 and Akt phosphorylation, as has been reported for IGF-1[36].

The slowing of OPC motility in *Pcdh15* knockdown cultures was associated with an accumulation of F-actin within the lamellipodia (Fig. 7). The cytoplasmic region of classic cadherin family members binds p120-catenin and β-catenin, which in turn binds α-catenin to interact with F-actin[86,87]. However, members of the protocadherin family, including Pcdh8, Pcdh10, Pcdh17, Pcdh18, Pcdh19, instead influence actin cytoskeletal dynamics by directly binding WASP family complexes[88]. As different members of the WASP family show discrete subcellular localisation, Pcdh15 may bind them directly to influence Arp2/3 activity. Our data suggest that in the lamellipodia, Pcdh15 affects WASP and Arp2/3 complex activity by modulating the activity of the upstream Cdc42 GTPase. Blocking the activity of Cdc42 or Arp2/3 was equally effective at reducing F-actin accumulation in the lamellipodia of Pcdh15 knockdown OPCs, and blocking Cdc42 activity rescued the contact time between filopodia and lamellipodial dynamics (veiling time) in Pcdh15 knockdown cultures (Fig. 9 and Supplementary Fig. 3). Previous studies have shown that Arp2/3 signalling is an important regulator of cells of the OL lineage and both OPCs and OLs express WASP family members and Arp2/3 proteins in their cell bodies and processes[12] (Fig. 8). OPCs cultured from *Arpc2* knock-out mice have a phenotype more severe than Pcdh15 knockdown OPCs, as they fail to produce lamellopodia, but also have slowed movement[89], and N-WASP inhibition reduces OPC filopodial number and length[12]. In addition, experiments including the conditional deletion of *Arpc3* from *CNP*[cre+] cells and the CK666 mediated blockade of Arp2/3 activity in immature OLs in vitro have shown that Arp2/3-mediated actin assembly is important for myelin initiation and axon engagement[44], making it intriguing to speculate that Pcdh15 could also facilitate this process.

Pcdh15 differentially regulates cytoskeletal dynamics at the lamellipodia and the soma—the site of process initiation, which requires microtubule assembly[66]. In cultured immature OLs, CNPase (2′,3′-cyclic nucleotide 3′-phosphodiesterase) binds tubulin heterodimers to drive microtubule assembly and support process elaboration[90]. Arp2/3 is critical for nucleation of the actin filament, however, ADF/Cofilin actively severs the actin filaments[91]. This process of actin treadmilling is critical to ensure a constant supply of actin monomers at the barbed ends, but the loss of the Cofilin homologue, Twinstar, suggests that it is also

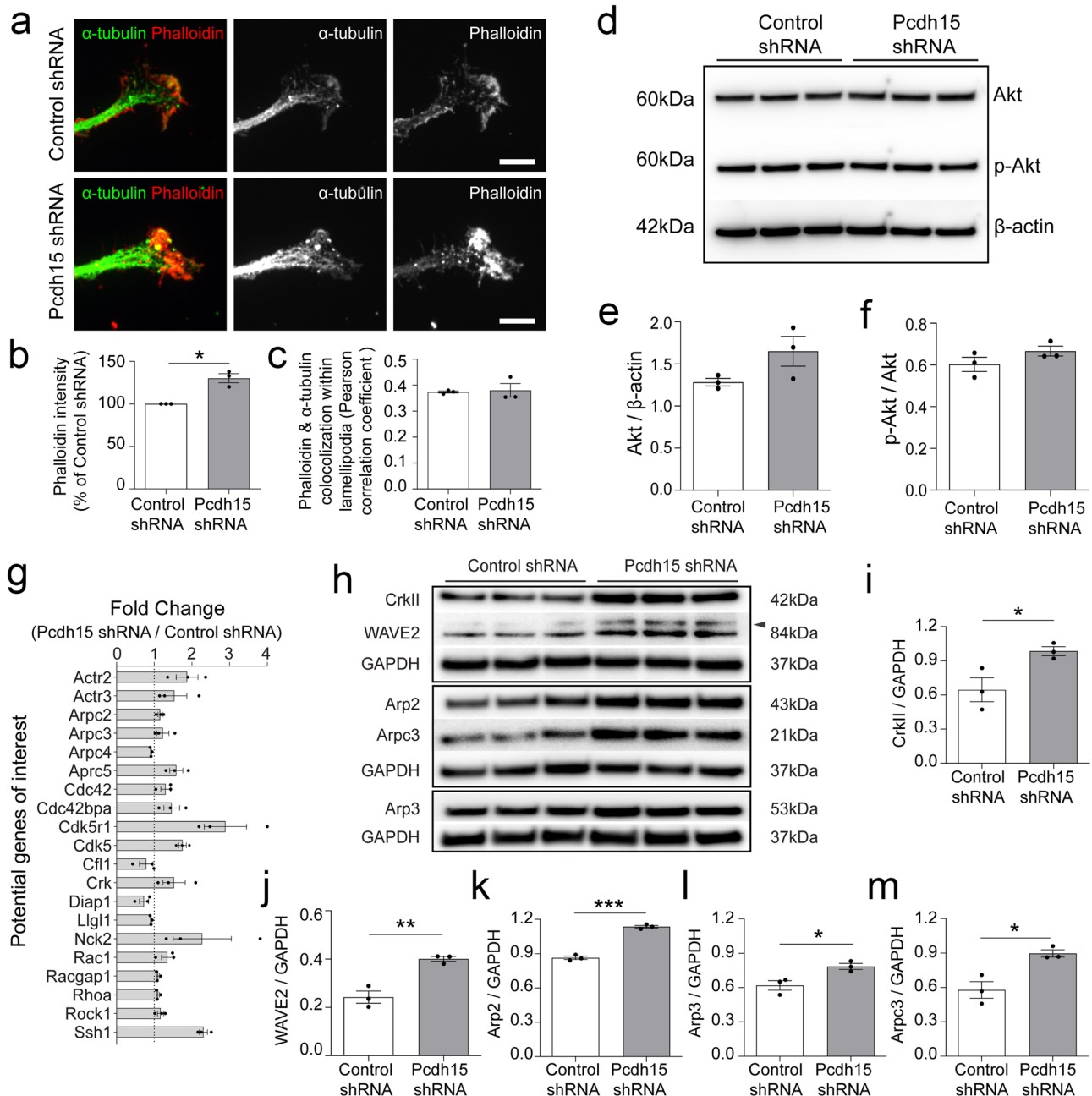

**Fig. 7 Pcdh15 knockdown promotes F-actin accumulation in OPC veils and increases the expression of genes and proteins that enhance Arp2/3 activity. a** Confocal images of OPC processes in control- and *Pcdh15* shRNA treated cultures immunolabelled to detect microtubules (α-tubulin, green) and F-actin (Phalloidin, red). **b** Quantification of phalloidin intensity (integrated pixel density) in control and Pcdh15-knockdown OPCs (normalized to the average intensity of control OPC cultures) ($n = 3$ independent cultures, ≥51 veils analysed per condition; one-sample $t$-test for Pcdh15-shRNA: deviation from 100(%) = 30.25, SD of deviation = 9.19, $t$ (2) = 5.697, $p = 0.029$). **c** Degree of colocalization between actin (phalloidin) and microtubule (α-actin) labelling in control and Pcdh15 knockdown OPC veils. Colocalization is expressed as a Pearson correlation coefficient (mean ± SEM, $n = 3$ independent cultures, ≥37 veils measured per condition; unpaired $t$-test, $p = 0.81$). **d** Protein lysates from control and Pcdh15 knockdown OPCs analysed by Western blot to detect Akt, phosphorylated Akt (p-Akt) and β-actin. **e** Expression of Akt in control and Pcdh15 knockdown OPC cultures, normalised to β-actin expression (integrated pixel density measured for each protein band; mean ± SEM, $n = 3$ independent cultures; unpaired $t$-test, $p = 0.12$). **f** Quantification of p-Akt (Ser473) and Akt in control and Pcdh15 knockdown OPCs, normalised to β-actin (integrated pixel density measured for each protein band; mean ± SEM, $n = 3$ independent cultures; unpaired $t$-test, $p = 0.20$). **g** Average fold change in cytoskeletal gene expression between OPCs transfected with control- and *Pcdh15*-shRNA (average from $n = 3$ independent cultures). **h** Protein lysates from control and Pcdh15 knockdown OPCs analysed by Western blot to detect CrkII and WAVE2, which are upstream of the Arp2/3 complex, Arp2, Arp3 and Arpc3, which are components of the Arp2/3 complex, and GAPDH. **i** CrkII, **j** WAVE2, **k** Arp2, **l** Arp3, and **m** Arpc3 protein expression was quantified by measuring the integrated pixel density of the corresponding Western blot band and normalising expression to GAPDH expression for each sample [mean ± SEM, $n = 3$ independent cultures, expression in control and Pcdh15 knockdown OPCs was compared by unpaired $t$-tests: **i** $p = 0.039$, **j** $p = 0.004$, **k** $p = 0.0001$, **l** $p = 0.028$, **m** $p = 0.016$]. Scale bars represent 5 μm.

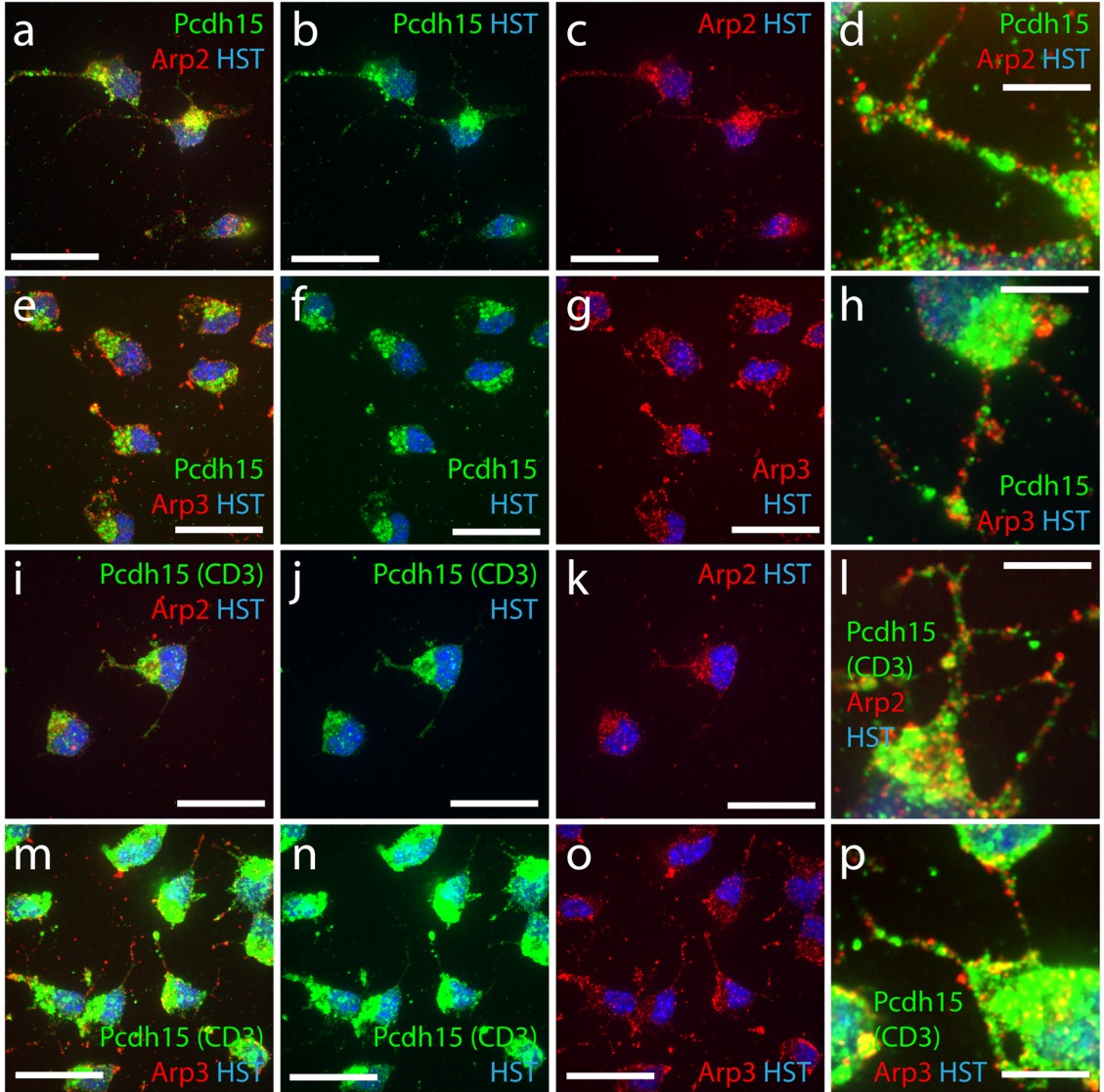

**Fig. 8 Primary OPCs express Arp2, Arp3 and Pcdh15 at the soma and throughout the processes. a–c** Single z-plane confocal scan of primary mouse OPCs processed to detect Arp2 (mouse anti-Arp2; red), Pcdh15 (rabbit anti-pan Pcdh15; green) and Hoechst 33342 (HST; blue). **d** Single confocal scan (100×) of primary mouse OPC processes expressing Arp2 (red) and Pcdh15 (green). **e–g** Single z-plane confocal scan of primary mouse OPCs processed to detect Arp3 (mouse anti-Arp3; red), Pcdh15 (rabbit anti-pan Pcdh15; green) and HST (blue). **h** Single z-plane confocal scan (100×) of primary mouse OPC processes expressing Arp3 (red) and Pcdh15 (green). **i–k** Single z-plane confocal scan of primary mouse OPCs processed to detect Arp2 (mouse anti-Arp2; red) and the CD3 isoforms of Pcdh15 (rabbit anti-CD3 Pcdh15; green) and HST (blue). **l** Single z-plane confocal scan (100×) of primary mouse OPC processes expressing Arp2 (red) and the CD3 isoforms of Pcdh15 (green). (**m-o**) Single z-plane confocal scan of primary mouse OPCs processed to detect Arp3 (mouse anti-Arp3; red), the CD3 isoforms of Pcdh15 (rabbit anti-CD3 Pcdh15; green) and HST (blue). **p** Single z-plane confocal scan (100×) of primary mouse OPC processes expressing Arp3 (red) and the CD3 isoforms of Pcdh15 (green). Scale bars represent 20 µm (**a–c**, **e–g**, **i–k**, **m–o**) or 5 µm (**d**, **h**, **l**, **p**).

critical for microtubule infiltration into the axon tips and branch points to facilitate axon branching[92]. Actin treadmilling is influenced by mTOR signalling, as the loss or inhibition of mTOR decreases ArpC3 expression and increases cofilin activity, but decreases OL branching complexity[48]. Genes encoding LIM kinase and Ssh1 phosphatase, enzymes that activate and deactivate Cofilin, respectively, are upregulated in Pcdh15 knockdown OPCs. If Pcdh15 knockdown OPCs experience a local reduction in cofilin activity at the soma, enhanced F-actin accumulation could impair process initiation and extension. Similarly, blocking Cdc42 in control OPCs could phenocopy the effect of Pcdh15 knockdown at the soma by decreasing MRCKα, LIM kinase and cofilin activity at the soma, leading to F-actin accumulation that

would hinder microtubule protrusion during process initiation and extension[92].

## Methods

**Transgenic mice**. All animal experiments were approved by the University of Tasmania Animal Ethics (A0013741, A0016151 and A0018606) and Institutional Biosafety Committees and were carried out in accordance with the Australian code of practice for the care and use of animals for scientific purposes. *Pdgfrα-H2BGFP* (stock # 007669) [*Pdgfrα-histGFP*[93,94]], *Sun1-eGFP* (stock # 021039)[95], *Ng2-CreER^T2* (stock # 008538)[8] and *PLP1-CreERT* (stock # 005975)[96] mouse lines were purchased from Jackson Laboratories. Postnatal day (P) 0-2 *Pdgfrα-histGFP* neonates were genotyped using a BlueStar flashlight (Nightsea, Lexington USA) to detect GFP expression in the brain, paws and ears. Mice expressing Cre-recombinase or eGFP were genotyped by PCR[97] using the following primer sequences: Cre 5′ CAGGT CTCAG GAGCT ATGTC CAATT TACTG ACCGTA

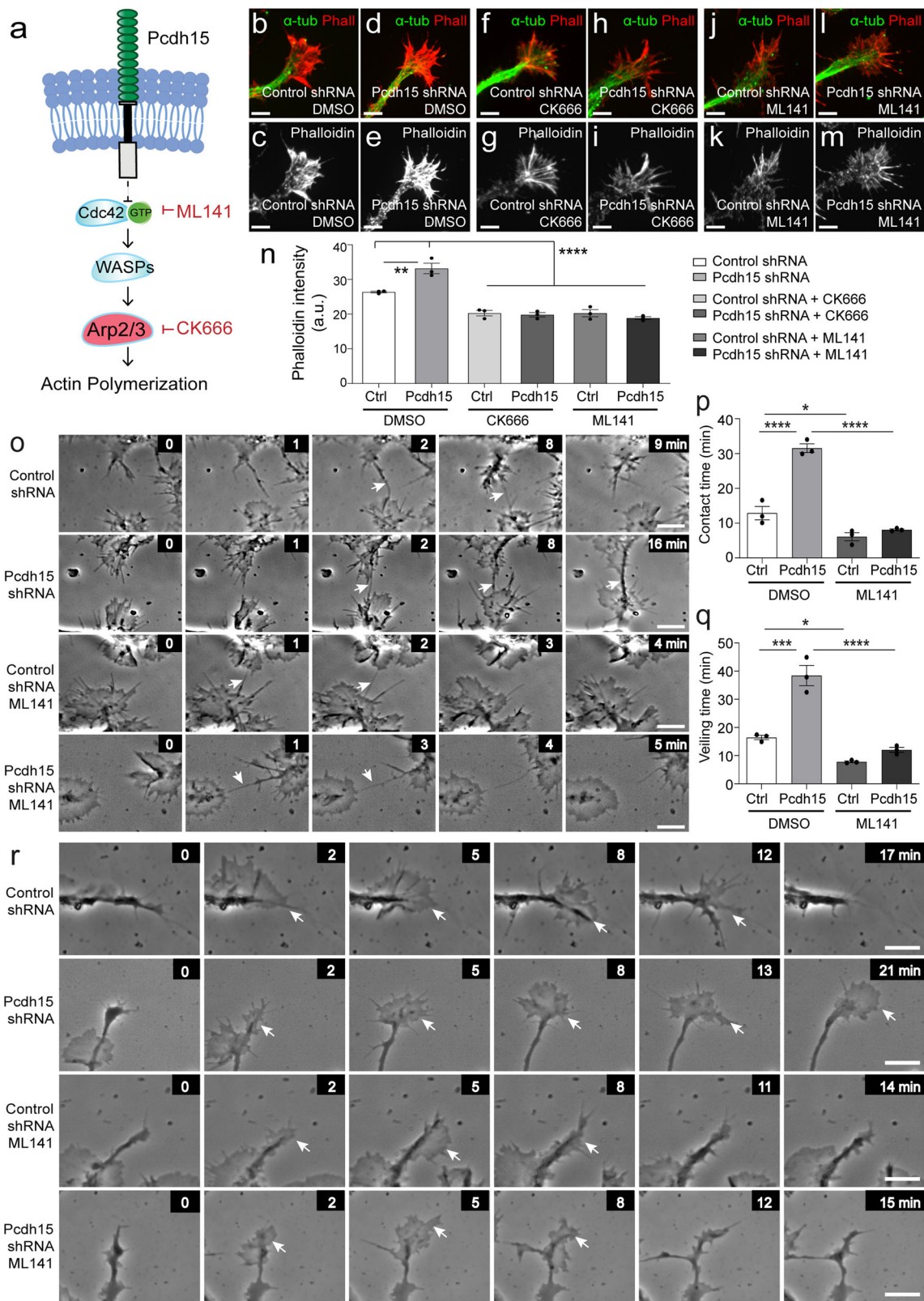

and Cre 3′ GGTGT TATAAG CAATCC CCAGAA; GFP 5′ CCCTG AAGTTC ATCTG CACCAC and GFP 3′ TTCTC GTTGG GGTCT TTGCTC. Mice were maintained on a C57BL/6 background and inter-crossed to generate male and female offspring for experimental use. Mice were weaned >P30 to ensure appropriate myelin development, were group housed with same-sex littermates in Optimice micro-isolator cages (Animal Care Systems, Colorado, USA), and were maintained on a 12 h light/dark cycle at 20 °C with uninhibited access to food and water.

**INTACT nuclear RNA isolations**. Cre-inducible *Sun1-eGFP* transgenic mice were crossed with *Ng2-CreER^T2* or *PLP1-CreERT* transgenic mice to generate *Ng2-CreER^T2:: Sun1-eGFP* or *PLP1-CreERT:: Sun1-eGFP* mice, respectively. Tamoxifen (20 mg/kg) was delivered to adult mice (18-22 weeks) daily for 5 consecutive days, to induce expression of the Sun1-eGFP transgene in NG2⁺ OPCs or PLP⁺ OLs. Brains were harvested two days later and Sun1-eGFP⁺ nuclei were affinity purified using anti-GFP (Abcam 13970) and magnetic beads (Dynabead)[95]. Nuclear RNA was converted to cDNA and libraries were generated using the Clonetech

**Fig. 9 Pcdh15 suppresses Cdc42 and Arp2/3 activity to promote filopodial repulsion and process veiling by OPCs. a** A schematic proposing a mechanism by which Pcdh15 could impact actin polymerisation in OPCs and depicting the point of action in the pathway of the drugs ML141 and CK666. **b–m** Confocal images of OPC processes in control- and *Pcdh15*-shRNA cultures, treated with diluent (DMSO), CK666 (50 μM) or ML141 (10 μM), and immunolabelled to detect microtubules (α-tubulin, green) and F-actin (Phalloidin, red). **n** Phalloidin intensity (integrated pixel density) in the veils of OPCs from control- and *Pcdh15*-shRNA cultures treated with DMSO, CK666 or ML141 [mean ± SEM, $n = 3$ independent cultures and $\geq 50$ veils analysed per treatment condition; two-way ANOVA: shRNA treatment $F_{(1,12)} = 5.22$, $p = 0.041$; drug treatment $F_{(2,12)} = 86.54$, $p < 0.0001$; interaction $F_{(2,12)} = 13.14$, $p = 0.0009$ with Bonferroni multiple comparisons, $*p < 0.01$, $****p < 0.0001$]. **o** Image series extracted from time-lapse videos depicting the filopodia of adjacent OPCs making contact in control- and *Pcdh15*-shRNA cultures treated with DMSO or ML141. Arrows indicate contacting filopodia. All images are time-stamped relative to the first image in the depicted series (designated as 0 min) and images are selected to highlight the formation and maintenance of filopodial contacts. **p** The mean duration that OPC filopodia remain in contact (min) in control- and *Pcdh15*-shRNA cultures treated with DMSO or ML141 [mean ± SEM, $n = 3$ independent cultures, $\geq 61$ filopodial contacts analysed per treatment condition; two-way ANOVA: shRNA treatment $F_{(1,8)} = 63.09$, $p < 0.0001$; drug treatment $F_{(1,8)} = 135.0$, $p < 0.0001$; interaction $F_{(1,8)} = 40.95$, $p < 0.0001$ with Bonferroni multiple comparisons, $*p < 0.05$, $****p < 0.0001$]. **q** The mean time taken for an OPC process to extrude and retract a veil in control- and *Pcdh15*-shRNA cultures treated with DMSO or ML141 [mean ± SEM, $n = 3$ independent cultures, $\geq 122$ veils quantified per treatment condition; two-way ANOVA: shRNA treatment $F_{(1,8)} = 47.69$, $p = 0.0001$; drug treatment $F_{(1,8)} = 84.96$, $p < 0.0001$; interaction $F_{(1,8)} = 21.94$, $p = 0.0016$ with Bonferroni multiple comparisons, $*p < 0.05$, $***p < 0.001$, $****p < 0.0001$]. **r** Image series selected from time-lapse videos of control- and *Pcdh15*-shRNA OPCs treated with DMSO or ML141, showing OPC processes as they extrude and retract veils. All images are time-stamped relative to the first image in the depicted series (designated as 0 min) and images are selected to highlight veil extrusion and retraction. Arrows indicate elaborated veils. Scale bars represent 2 μm (**b–m**) or 5 μm (**o, r**).

SMARTer stranded RNA-Seq kit. Libraries were sequenced using an Illumina HiSeq 2500. Genome alignment, STAR analysis, and DESeq differential expression analysis were performed using workflows on Basepair (https://www.basepairtech.com).

**Primary mixed glial cultures**. Primary mixed glial cultures were prepared from P0-5 *Pdgfrα-histGFP* mice as previously described[98]. Briefly, cortices were dissected into sterile Earle's Balanced Salt Solution (EBSS, Invitrogen, 14155–063), diced into small pieces and digested in 0.25% (w/v) trypsin (Sigma, T4799) in EBSS at 37 °C for 50 min before the addition of 10% foetal calf serum (FCS, Invitrogen, 10099–141) and 0.12 mg/ml DNAse 1 (Sigma, 5025). The cells were dissociated by gentle trituration, passed through a 40μm cell sieve (BD, 352,340) pelleted by centrifugation, and resuspended in complete OPC medium [20 ng/ml human PDGF-AA (Peprotech), 10 ng/ml bFGF (R&D Systems), 10 ng/ml human CNTF (Peprotech), 5 μg/ml NAC (Sigma), 1 ng/ml NT3 (Peprotech), 1 ng/ml biotin (Sigma), 10 μM forskolin (Sigma), 1x penicillin/streptomycin (Invitrogen), 2% B27 (Invitrogen), 50 μg/ml insulin (Sigma), 600 ng/ml progesterone (Sigma), 1 mg/ml transferrin (Sigma), 1 mg/ml BSA (Sigma), 400 ng/ml sodium selenite (Sigma) and 160μg/ml putrescine (Sigma) in DMEM + Glutamax (Invitrogen)]. The cells were plated across three wells of a six-well plate per cultured pup. Wells were pre-coated with >300,000 MW Poly-D-Lysine (PDL, Sigma, P7405), and were incubated at 37 °C with 5% $CO_2$.

**Isolating OPCs from primary glial cultures**. GFP⁺ OPCs were isolated from primary mixed glial cultures at 7 days in vitro (7DIV) by fluorescence-activated cell sorting (FACS)[99] or immunopanning[98]. In brief, cells were detached by exposure to 50% (v/v) TrypLE (Thermo Fisher Scientific, 12604013) in EBSS, the trypsin inactivated by the addition of FCS, and the cells resuspended in 50% (v/v) FCS in EBSS and passed through a 40 μm cell sieve. For FACS sorting, propidium iodide (PI, Thermo Fisher Scientific, P3566) was added to the cell suspension. GFP⁺ and PI-negative live OPCs were then purified by FACS using a Beckman Coulter MoFlo Astrios cell sorter (Beckman Coulter, CA, USA) as previously described[99]. For purification by immunopanning, the cell suspension was transferred to a petri dish coated with anti-PDGFRα (also known as CD140a; BD Pharmigen 558774; RRID:AB_397117) and incubated for 45 min to allow OPCs to adhere to the plate[100]. Non-adherent cells were then removed with an EBSS wash and the purified OPCs stripped by a 5-min incubation with TrypLE™ Express (ThermoFisher, 12604013) diluted 1:5 with EBSS. For western blot analysis, $4 \times 10^4$ OPCs were plated into 12-well plates. For immunocytochemistry, $3 \times 10^4$ OPCs were plated onto 18 mm glass coverslips placed in 24-well plates. For live imaging, purified OPCs were cultured on 13 mm glass coverslips at a density of $4 \times 10^4$ cells per well in 12-well plates. All OPC cultures were incubated at 37 °C and 5% $CO_2$ in OPC complete medium.

**OPC transfection with shRNA lentiviral particles**. For each independent culture, one or more wells of the culture plate were randomly assigned to each experimental group. Care was taken to ensure that each treatment group was represented across each independent culture. OPC transfections were carried out at 9 days in vitro (2 days after re-plating). OPC complete culture medium was replaced with OPC complete culture medium containing 5 μg/ml polybrene (Santa Cruz Biotechnology, sc-132220) and the relevant shRNA lentiviral particles: control shRNA lentiviral particles-A (Santa Cruz Biotechnology, sc-108080) or *Pcdh15* shRNA lentiviral particles (Santa Cruz Biotechnology, sc-152056-V), which contained a mixture of: *Pcdh15* shRNA-A (sc-152056-VA: GATCC GGATA AGACT CGCTA CTATT TCAAG AGAAT AGTAG CGAGT CTTAT CCTTT TT); *Pcdh15* shRNA-

B (sc-152056-VB: GATCC GTCTG CACAT CGAAA TACTT TCAAG AGAAG TATTT CGATG TGCAG ACTTT TT) and *Pcdh15* shRNA-C (sc-152056-VC: GATCC GACTA TGCCA CCTGG TATAT TCAAG AGATA TACCA GGTGG CATAG TCTTT TT). After 12 h at 37 °C/5% $CO_2$, the shRNA-containing polybrene medium was replaced with OPC complete culture medium.

**Delivery of pharmacological agents to OPCs**. The MEK inhibitor (U0126, Promega, V1121), Arp2/3 inhibitor (CK666, Millipore, 182515) and Cdc42 inhibitor (ML141, Sigma, SML0407) were reconstituted to a concentration of 4.3 mg/mL (10 mM), 30 mg/mL (100 mM) and 4.1 mg/mL (10 mM), respectively, in dimethyl sulfoxide (DMSO, Sigma, 472301) and stored at −20 °C. DMSO, U0126 (10 μM), Ck666 (50 μM) or ML141 (10 μM) were added to the polybrene transfection medium containing either control- or *Pcdh15*-shRNA, as described above. Note that DMSO or 10 μM U0126 in OPC complete medium was also added 10 min before lentiviral transfection. For EdU experiments, OPCs were exposed to each pharmacological agent for a further 12 h, in the presence of EdU. For live imaging experiments, the relevant pharmacological agent was also included in the live imaging buffer (see below).

**Time-lapse microscopy and quantification of OPC motility**. Even in the absence of a chemotactic or chemorepulsive gradient, OPC processes continually extrude and retract lamellipodia-like structures[101,102]. To examine this form of OPC motility, OPC cultures (12 h post-transfection) were transferred to the heated stage (37 °C) of an enclosed Nikon Eclipse TI microscope (Tokyo, Japan) with a 488-emission filter, and time-lapse video recordings were made using a 40× air or 100× oil immersion objective, with movies recorded from ≥5 regions of interest per coverslip. During imaging, cultures were continually perfused with live imaging buffer [10% HBSS/15 mM HEPES (Sigma)/10 mM Glucose/2 mM $CaCl_2$/1 mM $MgCl_2$/1% B27/ MilliQ water, PH 7.4], with or without DMSO, U0126 (10 μM) or ML141 (10 μM). Phase images were acquired every 1 min for 2 h using an Andor Zyla camera (Andor Technology, Belfast, Northern Ireland). All analyses were carried out by an experimenter blind to the treatment conditions, and time-lapse movies were opened and analysed in Fiji/ImageJ (NIH, USA) to quantify: veiling time (time for a process to complete a single extrusion and retraction of its lamellipodia-like veil); the duration of an individual filopodial contact made between adjacent OPCs, and the number of new processes elaborated by individual OPCs per hour of imaging. Quantification was performed by manually scoring events as present or absent across every video frame (1 min apart) to calculate event duration. To quantify process number per OPC, still images were collected from each time-lapse series at a single time point (1 min into the imaging series). Each still image was opened in image J and used to manually quantify the number of processes supported by individual OPCs within each field of view.

The basal motility of cultured OPCs was quantified from time-lapse video files containing images collected every min for 2 h. The video files were opened in ImageJ (National Institutes of Health, USA) and tracing of the cell soma performed using the Manual Tracking plugin (Fabrice Cordelières, Institut Curie, Orsay, France). OPC motility is minimal under these cultural conditions. However, OPCs were selected by researchers who were blind to the culture conditions and assigned for motility tracing if they were within the central third of the imaging field at the onset of image collection (reduced the likelihood of the cell moving outside of the imaging field) and were near the edge or outside of an OPC cluster. Each track was analysed using the Chemotaxis and Migration Tool 2.0 (Ibidi GmbH, Munich, Germany). The software was also used to generate trajectory plots, calculate speed,

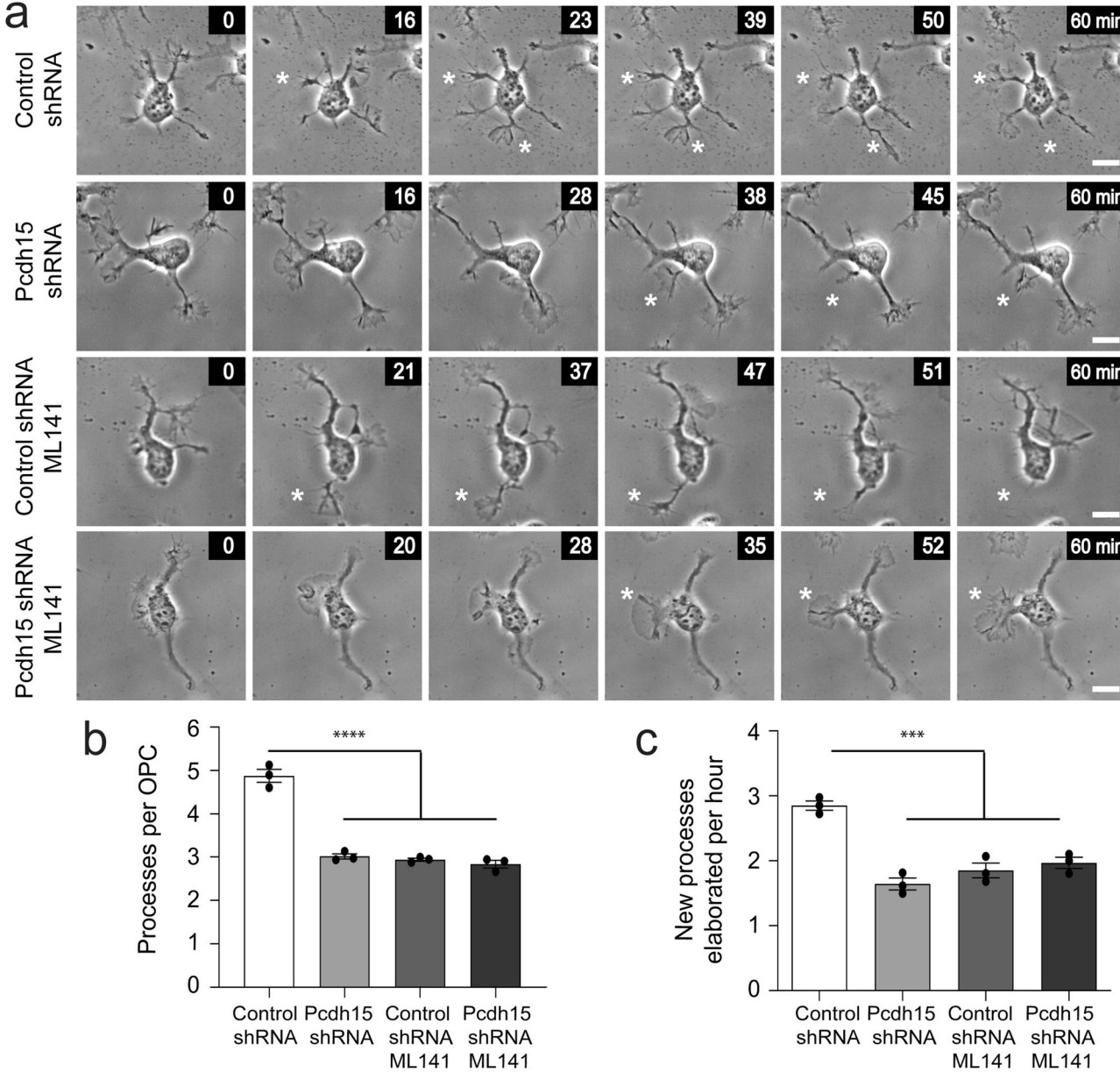

**Fig. 10 Inhibiting Cdc42 activity phenocopies the effect of Pcdh15 knockdown on OPC process number and generation. a** Image series extracted from time-lapse videos to show individual control and Pcdh15 knockdown OPCs treated with DMSO or ML141 (10 µM), as they elaborate new processes over time. All images are time-stamped relative to the first image in the depicted series (designated as 0 min). White asterisks indicate the new process. **b** The mean number of processes supported by control and Pcdh15 knockdown OPCs treated with DMSO or ML141 [mean ± SEM, $n = 3$ independent cultures, $\geq 76$ OPCs analysed per condition; one-way ANOVA: $F(3,8) = 108.4$, $****p < 0.0001$]. **c** The mean number of new processes that OPCs elaborated per hour in control and Pcdh15 knockdown cultures treated with DMSO or ML141 [mean ± SEM, $n = 3$ independent cultures and $n \geq 34$ cells analysed per culture condition; two-way ANOVA: shRNA treatment $F(1,8) = 13.37$, $p = 0.0064$; drug treatment $F(1,8) = 35.06$, $p = 0.0004$; interaction $F(1,8) = 51.65$, $p < 0.0001$ with Bonferroni multiple comparisons, $***p < 0.001$]. Scale bars represent 10 µm.

and quantify the total distance moved and the Euclidean distance (distance from start to end point as a direct line).

**Generating protein lysates and performing western blots**. Protein lysates were generated from cultured OPCs at 12 days in vitro (5 days after re-plating). OPC cultures were washed thrice with cold phosphate-buffered saline (PBS) before the application of lysis buffer (50 mM Tris-HCL, 150 mM NaCl, 1% NP-40, 1% sodium deoxycholate, 0.1% SDS and one phosphatase inhibitor tablet). OPCs were dislodged using a cell scraper and collected into a 15 mL tube for trituration. Protein lysates were also generated from adult mouse brains following transcardial perfusion with ice-cold PBS. The brain was removed onto an ice-cold petri-dish and sliced into 2 mm coronal slices using a brain matrix (Agar Scientific). Each coronal slice was homogenized separately in an ice-cold lysis buffer. Lysates were centrifuged at 13,000 rpm for 10 min at 4 °C and the supernatant was stored at −80 °C.

The protein concentration of each lysate was quantified by Bradford Assay, using a Pierce BCA Protein Assay Kit (Thermo Scientific) with bovine serum albumin (BSA; Sigma A7030) standards according to the manufacturer's instructions. Briefly, 5 µl of each BSA standard or protein lysate was loaded into a 96-well plate in triplicate. 25 µl of the assay solution, comprising 1 ml of Reagent A (Bio-Rad, 5000113) and 20 µl of Reagent S (Bio-Rad, 5000115), was added to each well, followed by 200 µl of Reagent B (Bio-Rad, 5000114). After 15 min, the plates were analysed using a FLUOstar OPTIMA microplate reader. Protein concentrations were plotted against absorbance for each standard and used to calculate the protein concentration of each sample.

Protein expression was quantified by western blot as previously described[103]. Briefly, 10 µg of each protein lysate was mixed with 4×Blot LDS sample buffer (Invitrogen, B007), 500 mM DL-Dithiothreitol (DTT; Sigma, D0632) and MilliQ water, and incubated at 70 °C for 10 min. Five microlitres of protein ladder (Invitrogen, LC5925) or a protein sample was loaded into each well of a 4–12% Bis-

Tris gel (Invitrogen, NW04120BOX) in MES running buffer (5% Blot running buffer/ MilliQ water) and run for 40 min at 166 V. The protein was transferred from the gel onto polyvinylidene BlotTM fluoride (PVDF) membrane (Bio-Rad, 162–0177) in transfer buffer (5% Bolt transfer buffer/10% methanol/0.1% Blot antioxidant/ MilliQ water), by performing electrophoresis (20 V for 60 min). The PVDF membrane was subsequently blocked with 5% (w/v) powdered milk in 0.05% (v/v) Tween-20 in tris buffered saline (TBS) and incubated overnight at 4 °C with the following primary antibodies: rabbit-anti-pan Pcdh15 (HL5614, 1:2000[29]), mouse-anti-p-ERK (Abcam; 1:1000), rabbit-anti-ERK (Abcam; 1:1000), rabbit-anti-p(Ser473)-AKT (Cell Signalling Technology; 1:1000), rabbit-anti-AKT (Cell Signalling Technology; 1:1000), rabbit-anti-p(ser37/37/Thr41)-β-catenin (Cell Signalling Technology; 1:1000), rabbit-anti-β-catenin (Cell Signalling Technology; 1:1000), mouse-anti-WAVE2 (Santa Cruz Biotechnology; 1:100), mouse-anti-Arp2 (Santa Cruz Biotechnology; 1:100), mouse-anti-Arp3 (Santa Cruz Biotechnology; 1:100), mouse-anti-p21-Arc (anti-Arpc3; BD Transduction Laboratories; 1:1000), or mouse-anti-CrkII (BD Transduction Laboratories; 1:1000). PVDF membranes were washed twice in TBS-T before being incubated with the horseradish peroxidase (HRP) conjugated secondary antibodies [goat anti-mouse HRP (1:20,000, P044701-2, Dako) or goat anti-rabbit HRP (1:20,000, P044801-2, Dako)] diluted in 1% (w/v) powdered milk in TBS-T at 37 °C for 1 h. Antibody labelling was detected using a chemiluminescence system (ECL; Millipore, WBKLS0500) and imaged using an Amersham imager 600 (GE Healthcare, Chicago, IL). When multiple proteins were analysed from a single gel, stripping was carried out using Restore plus western blot stripping buffer (Thermo Scientific, USA) according to the manufacturer's instructions. In particular, the expression of the control proteins GAPDH (EMD Millipore, 1:1000) and/or β-actin (Sigma, 1:1000) were quantified to allow protein expression to be standardised between samples. For western blot images shown in the main figures, the unedited western blot images are shown in Supplementary Fig. 4.

**RT²-profiler PCR arrays**. OPCs cultured in 12-well plates were washed with PBS before being incubated with 0.25% (w/v) trypsin (Sigma, T4799) in PBS at 37 °C for 5–8 min. Trypsin was inactivated with FCS, and the cells were pelleted by centrifugation before being lysed with QIAzol lysis buffer (QIAGEN, 79306). RNA was extracted and purified using a miRNeasy mini kit (QIAGEN, 217004) according to the manufacturer's instructions. The concentration of RNA in each sample was determined using a NanoDropTM spectrophotometer (Thermo Fisher Scientific, USA) and the quality of the RNA evaluated by running 1 μg of each sample on a 1% (w/v) agarose/1% (w/v) sodium hypochlorite (Sigma, 239305) gel at 100 V for 30 min. cDNA was synthesised using the RT2 First Strand Kit (Qiagen, 330401) according to the manufacturer's instructions. Genomic DNA was eliminated by combining 1 μg of RNA with 2 μl of Buffer GE, and the samples were incubated at 42 °C for 5 min before being transferred to ice for 1 min. Samples were then mixed 1:1 with the reverse-transcription mix (40% 5x Buffer BC3/10% control P2/20% RE3 reverse transcriptase mix/30% RNase-free water) and incubated at 42 °C for 15 min before being transferred to 95 °C for 5 min. The resulting cDNA was diluted in RNase-free water and stored at −20 °C.

A quantitative PCR array was performed to examine the expression of mouse cytoskeletal genes by OPCs. We utilised the RT² SYBR Green qPCR Mastermix (Qiagen, 330501) and RT² ProfilerTM PCR array of mouse cytoskeleton regulators (Qiagen, PAMM-088Z-6), according to the manufacturer's instructions. Each 96-well array contained primers for 84 cytoskeletal genes, 3 reverse transcription controls, 3 positive PCR controls and 5 housekeeping genes including *Actb*, *B2m*, *Gapdh*, *Gusb* and *Hsp90ab1*. Each cDNA reaction was mixed 1:1 with the 2x RT² SYBR Green qPCR Mastermix, 25 μl was loaded per well of the RT² ProfilerTM PCR array, and the DNA amplified in a Roche LightCycler 480 using the following program: 95 °C for 10 min, and 45 cycles of 95 °C for 15" and 60 °C for 1 min. This assay was performed on cDNA from $n = 3$ independent cultures and threshold cycle (Ct) values for *Gapdh* were used to normalize gene expression. Gene expression comparisons were performed using Qiagen software (http://dataanalysis.qiagen.com/pcr/arrayanalysis.php).

**EdU administration**. To label dividing cells in vitro, 5-ethynyl-2′-deoxyuridine (EdU, Thermo Fisher Scientific, E10415) was reconstituted to 2.5 mg/ml in EBSS, filter sterilised and stored at −20 °C. OPCs (12 h post-transfection) were incubated in complete culture medium containing 2.5 μg/ml EdU for 12 h at 37 °C/5% CO₂ before immersion fixation in 4% (w/v) PFA in PBS for 20 min at ~21 °C. EdU was detected using the Alexa Fluor azide 647 Click-IT EdU Imaging kit (Invitrogen) according to the manufacturer's protocol.

**Tissue preparation and freezing**. Embryonic day (E) 16.5 mice were decapitated, the brains and spinal cords removed, and immersion fixed in 4% (w/v) PFA in diethyl pyrocarbonate (DEPC)-treated PBS overnight at 4 °C, before being cryo-protected in 20% (w/v) sucrose in DEPC-treated PBS. Tissues were snap frozen in Cryomatrix (Shandon UK), using liquid nitrogen, and stored at −80 °C.

**RNA probe synthesis and in situ hybridisation**. To investigate the expression of *Pdgfrα* and *Pcdh15* in the E16.5 mouse spinal cord, RNA probes were synthesised from linearised DNA templates (Pcdh15: A730051L15; Pdgfrα: 1.5 kb SacI-PvuII

fragment encompassing most of the extracellular domain of the rat PDGFRα, cloned into pGEM1). For each riboprobe synthesis reaction, 5 μl of DNA template was mixed with 5 μl of 5× transcription buffer (Promega, P1181), 7.5 μl of DTT (Sigma, D0632), 2.5 μl of digoxin (DIG)-labelled deoxynucleotides (Roche, 11277073910, Switzerland), 1 μl of RNAse inhibitor (Promega, N2611), 2 μl of T7 RNA polymerase for the *Pdgfra* probe or T3 RNA polymerase for the *Pcdh15* probe (New England Biolabs, USA), and 2 μl of DEPC-treated water. The probe synthesis reaction was incubated at 37 °C for 2 h. Seventy-five microlitres of DEPC-treated water was added and the quality of the probe verified by running 4 μl on a 1% (w/v) agarose gel at 100 V for 15 min. RNA probes were aliquoted on ice and stored at −80 °C.

For in situ hybridisation, 25 μm E16 transverse spinal cord cryosections were collected onto glass slides and air dried. RNA probes were diluted 1:1000 in 65 °C hybridization solution [50% formamide/10% dextran sulfate/0.1% SDS/0.1 mg/ml Yeast tRNA (Roche)/1 × Denhardt's solution/ × hybridization salts solution (0.2 M NaCl, 5 mM EDTA, 10 mM Tris-HCl pH 7.5, 5 mM NaH2PO₄, 5 mM Na2HPO₄)] and denatured at 70 °C for 5 min. 200 μl of probe solution was applied to each slide before they were covered with a glass coverslip and incubated in a sealed humidified chamber overnight at 65 °C. Slides were thrice washed in 65 °C wash solution (50% formamide/1 × hybridization salts solution/0.1% Tween-20) and twice washed in maleic acid buffer-tween 20 (MABT; 100 mM maleic acid, 150 mM NaCl, 0.1% Tween-20), before blocking [2% blocking reagent (Sigma)/10% heat-inactivated sheep serum (Invitrogen) in DEPC-treated PBS] for 1 h at 21 °C. Alkaline phosphatase (AP)-conjugated anti-DIG Fab fragments (Roche, 11093274910) were diluted at 1:1500 in a blocking solution and applied to slides overnight at 4 °C. Slides were washed thrice in MABT, once in pre-stain buffer (100 mM NaCl/50 mM MgCl₂/100 mM Tris pH 9.5/0.1% Tween-20) and developed in staining buffer [1 × pre-stain buffer/0.1% nitroblue tetrazolium (NBT; Roche)/0.1% 5-bromo-4-chloro-3-indolyl-phosphate (BCIP; Roche)/10% PVA] at 37 °C until the blue chromogenic signal was visible (2–5 h). Sections were ethanol dehydrated and xylene treated before DPX mounting (Sigma, 44,581). Images were collected using a Zeiss Axio Lab.A1 microscope (Zeiss, Germany) with a N-Achroplan 5× objective (Zeiss, Germany). Images were opened in Image J and labelled cells manually counted within the white matter. White matter area was quantified in the x-y plane (in mm²), and the cell densities reported correspond to the number of labelled cells identified within that x-y area.

**Immunocytochemistry**. Cultured cells were fixed by immersion in 4% (w/v) PFA in PBS overnight at 4 °C and PBS washed before primary antibodies were applied in blocking solution (10% FCS/0.2% Triton X-100 in PBS) overnight at 4 °C. After PBS washing, Alexa Fluor-conjugated secondary antibodies, diluted in blocking solution, were applied overnight at 4 °C. Coverslips were washed thrice and mounted with DAKO fluorescent mounting medium (Sigma). Primary antibodies included: rabbit anti-Pcdh15[32] [0.45 mg/ml; 1:200; Figs. 1 and 2], rabbit-anti-pan Pcdh15 (HL5614[29], 1:100; Fig. 8), rabbit anti-CD3 Pcdh15 (PB375[29], 1:100; Fig. 8), rat anti-GFP (Nacalai Tesque, 1:2000), goat-anti-PDGFRa (R&D System; 1:100), mouse-anti-Arp2 (Santa Cruz Biotechnology SC376698; 1:10) and mouse-anti-Arp3 (Santa Cruz Biotechnology SC48344; 1:10). Secondary antibodies included: donkey-anti-rat 488 (Invitrogen; 1:1000), donkey-anti-rabbit 568 (Invitrogen; 1:1000), donkey-anti-rabbit 647 (Invitrogen; 1:1000), donkey-anti-rabbit 488 (Invitrogen; 1:1000), donkey-anti-mouse 647 (Invitrogen; 1:1000) and donkey anti-goat 647 (Invitrogen; 1:1000). Filamentous (F)-actin was also detected by exposing cells to Alexafluor-568 conjugated phalloidin (Invitrogen; 1:50) and cell nuclei were detected by exposing cells to Hoechst 33342 (Thermo Fisher Scientific; 1:1000).

**Confocal image collection and analysis**. Confocal images were obtained using an UltraView Nikon Ti Microscope with Volocity Software (Perkin Elmer, Massachusetts, USA). To quantify the proportion (%) of OPCs that express EdU, 20× compressed confocal images were collected from nine locations spanning each coverslip. EdU is incorporated into the DNA of cells as they transition through the S-phase of the cell cycle, however, EdU labelling can result in a mixture of strongly and faintly labelled cells. OPCs will be faintly labelled if, for example, they are already part way through S-phase when EdU labelling commences or have not completed S-phase when EdU labelling concludes. Conversely, EdU labelling can be very strong for OPCs that divide more than once during the labelling period. Each confocal image was opened in ImageJ and the GFP⁺ OPCs manually evaluated and classified as EdU⁺ if any EdU-labelling was detected in the nucleus or EdU-negative, and these numbers were used to calculate the proportion of EdU-labelled OPCs (EdU⁺ GFP⁺/GFP⁺ × 100). To calculate the average distance between EdU⁺ nuclei, each image was converted to a 16-bit image, inverted, and a threshold was set to ensure that all EdU⁺ nuclei were detected, and the Euclidean distance (EDM nuclei) was measured in ImageJ.

To quantify Pcdh15 expression in OPCs, 40× images were collected from six locations spanning each coverslip. The confocal imaging parameters were fixed for each experiment (equivalent for all biological replicas). A control shRNA coverslip was used to set the laser power, exposure time and gain to ensure clear visualisation of the immunocytochemical signal. Those settings were then applied to collect images from all coverslips related to that experiment. Imaging parameters applied to collect the images used for quantification in Fig. 2g were the same but were not equivalent to those used to collect images for Fig. 2l. Each image was opened in

ImageJ and converted to an 8-bit image. The custom shape tool was used to trace the full outline of each OPC (soma and processes) and the integrated density measured for this region of interest. PDGFRα labelling was instead quantified by outlining the soma as the region of interest and measuring the mean grey value (sum of grey values for all pixels in the selection/total number of pixels). To quantify F-actin expression and the colocalization of Phalloidin (F-actin) and α-tubulin (microtubule) within OPC veils, images were collected using a 100× (oil) objective from a minimum of 12 areas spanning each coverslip. OPC veils were selected as the region of interest and pixel integrated density was measured using Image J (Fiji).

**Analysis of Pcdh15 mRNA expression using a published data set**. We evaluated *Pcdh15* expression in OPCs and oligodendrocytes using a data set accessed through the Brain RNAseq web portal: http://web.stanford.edu/group/barres_lab/brain_rnaseq.html[15]. This RNA-seq data is available through the National Center for Biotechnology Information (NCBI) Gene Expression Omnibus (GEO) and is accessible through GEO Series accession number GSE52564.

**Statistics and reproducibility**. Data are presented as the mean ± standard error of the mean (SEM) for data collected from ≥3 independent cultures per condition. The sample size was not predetermined. The exact number of animals or cultures analysed (*n*) is stated in each figure legend and represents the number of independent biological replicates. No data were excluded from the analysis and detail of the statistical approach is also provided in each figure legend. Data were first assessed using a Shapiro-Wilk or Kolmogorov-Smirnov normality test and analysed by parametric or non-parametric tests as appropriate. Data sets with $n = 3$ in any group were analysed using parametric tests, as the nonparametric equivalents rely on ranking and are unreliable for small sample sizes (GraphPad Prism 8.0). Data comparing protein expression to normalised control values were analysed using a two-sided one-sample *t*-test in which the theoretical mean was set to 100%. Normally distributed data comparing two groups were analysed using two-sided unpaired *t*-tests, or by one-way ANOVA with Tukey's corrections for multiple comparisons when comparing three or more groups. Data with two independent variables (eg. shRNA and drug treatment) were analysed by two-way ANOVA with a Bonferroni multiple comparisons test. PDGFRα expression was compared between control and Pcdh15 knockdown OPCs using a Mann Whitney U test. All statistical analyses were performed using GraphPad Prism version 9 (GraphPad Software Inc., La Jolla, USA) and data sets were considered to be significantly different at $p < 0.05$. For the graphs presented in the main figures, the source data is available in Supplementary Data 1.

Please note that key morphological and behavioural differences identified between control- and Pcdh15-shRNA treated groups were also validated by reproduction across distinct experimental cohorts. In particular, an effect of Pcdh15 knockdown on: EdU labelling was detected in separate studies reported in Figs. 3g, h and 4l; filopodial contact time was detected in separate studies reported in Figs. 5a, 6b and 9p; veiling time was detected in separate studies reported in Figs. 5d, 6c and 9q; process number was detected in separate studies reported in Figs. 5f and 10b, and new process generation were detected in separate studies reported in Figs. 5h, 6f and 10c.

## Data availability

The use of all transgenic mouse lines complies with the conditions of their materials transfer agreements. Individual data points are displayed on the graphs and all source data underlying the graphs are available in Supplementary Data 1. The full western blot images corresponding to the image excerpts included in the main figures are included as Supplementary Fig. 4. The associated raw video and image files are a research resource that will be made available by the corresponding author, upon reasonable request.

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

## Acknowledgements

We thank our colleagues at the University of Tasmania for constructive feedback and suggestions for improvement. We thank Dominic Cosgrove for providing the rabbit anti-Pcdh15 antibody for immunocytochemistry (Boys Town National Research Hospital, Boys Town, United States of America, NE 68010). This research was supported by grants from the National Health and Medical Research Council of Australia (NHMRC: 1077792, 1139041), the Medical Research Future Fund (EPCD00008), MS Research Australia (11-014, 16-105, 17-007), the National Institute of Health (1R21NS111211-01A1, R01DC016295) and the Australian Research Council (DP180101494). Y.Z. was supported by scholarships from the Menzies Institute for Medical Research at the University of Tasmania and Anhui Medical University. B.S.S. was supported by a scholarship from Dementia Australia (K0025637). B.E. was supported by an endowment from the Warren family. A.Y.F. was supported by a Howard Hughes Medical Institute Gilliam Fellowship. C.L.C. was supported by a fellowship from MS Research Australia and the Penn Foundation (15-054). K.M.Y. was supported by a Paired Fellowship from MS Research Australia and the Macquarie Group Foundation (17-0223) and a Senior Research Fellowship from MS Australia (21-3023). The funding sources had no role in the study design; collection, analysis or interpretation of the data; writing the manuscript, or the publication process.

## Author contributions

Y.Z., R.G., C.L.C. and K.M.Y. developed the project. Z.M.A. and S.R. developed and provided research tools for this project. Y.Z., R.G., C.L.C., R.R., K.M.Y., B.S.S., B.E. and A.Y.F. carried out the experiments. K.M.Y., C.L.C., B.E. and Z.M.A. obtained the funding. Y.Z., B.E., R.R., R.G., K.M.Y. and C.L.C. performed the statistical analyses and generated the figures. K.M.Y., R.G., Z.M.A., B.E. and C.L.C. provided supervision. Y.Z. and K.M.Y. drafted, and B.E., Z.M.A. and R.G. edited the manuscript.

## Competing interests

The authors declare no competing interests.
