## [Peer Review File · Communications Biology]

Reviewers' comments:

Reviewer #1 (Remarks to the Author):

The manuscript by Dr. Kaylene Young group is quite interesting and very much informative in the field of glial biology in general and oligodendrocyte and myelin biology in particular.

Major claims of the paper are as follows:

- Pcdh15 regulate differential signalling at cell body and at cell periphery which is quite an interesting area to further explore.
- Pcdh15 signalling in OPCs play major role in maintaining the progenitor population in the brain and data from OPC studies also has implications for understanding glioma progression.
- This study is also first to report detailed understanding of role of Pcdh15 on lamellipodial and filopodial dynamics that are essential machinery for Oligodendrocyte progenitor migration and hence the current observations has lot of implications for demyelinating disorders in particular and neurodegenerative disorders in general.

Novelty aspects and Significance of the study:

- Wei Huang et al., 2020 Cell J, shows the role of PCDH15 cell repulsion and cell proliferation mechanisms.

However, there are just few studies detailing and further exploring the role and involvement of PCDH15 in OPC biology.

- Han et al., 2020 had reported that PCDH15 as direct target of miR22 and dysregulation of wnt and b-catenin pathways and thereby reduced cell proliferation.

Methodology and statistical analysis section of the manuscript:

- In Fig2 B,C the reduction in Pcdh15 protein level is very significant and has reduced almost 80-90 % . However, the shRNAs (A,B,C) used independently do not show that much reduction in the Pcdh15 levels in terms of pixel density. Authors may explain or justify the observations.
- In Fig3, the Edu+/GFP+ cells are not much different in the control and treated groups. Authors do mention that there were 3 independent experiments, however, there is no mention of how many different areas on the slides were scored for counting the Edu+/GFP+ cells to show the robustness of the observation.
- In Fig5F, how one can confirm that the processes shown are the new ones or the old ones. What is the methodology to follow the new process formation versus the existing ones so as to comment whether there is significant reduction in the number of new processes formed.
- In Fig 7D, total and phosphorylated Akt levels do not show any changes in either control or treated groups, however, the authors claimed that Pcdh15 has significant role in cell proliferation. The authors have not discussed about the potential role of Pcdh15 in regulating OPC marker proteins like PDGFRa or PDGFRb levels and their contribution in cell behaviour (proliferation/migration), which will be an important aspect as far as OPC behaviour biology is concerned.

Reviewer #2 (Remarks to the Author):

Zhen and colleagues present a paper entitled "Protocadherin 15 suppresses oligodendrocyte progenitor cell proliferation and motility via distinct signalling pathways."

In general, the paper is quite organized and presents interesting aspects of the transmembrane protein protocadherin 15 (Pcdh15) as a critical regulator of basal proliferation and process motility, important peculiarities that characterize the health of oligodendrocytes in the CNS. Still, at the same time, they identify the role of Pcdh15 in the case of glioma progression.

Each section of the paper (introduction, methodology, results, and statistical analysis) is clearly described. The discussion itself argues the data presented broadly. However, some aspects need to be clarified.

- How have the choice of concentrations of the various pharmacological agents been evaluated? Have you, by chance, considered any toxic effects on your oligodendrocyte cultures? Cell images do not always show cells in "health" after different treatments.
- U0126, do you use it at 16 microM or 20microM? In the text, it is indicated 16 and in the legend of figure 20microM. Which is the right one?
- In the legend of Fig 1 corresponding to panel D, what does "Expression data from 15" mean? Furthermore, the bar indicated in the legend for panels A-C is not present in the figure.
- In Fig 4A, the images of the bands of 5 samples (3 + 2) are shown, but then the densitometries are shown on 3 + 3. For clarity, perhaps it would be appropriate also to show the third sample for Pcdh15 shRNA.
- The legend of the exact figure is described in a somewhat confusing way. Could you please clarify the description of what is shown in the figure? For example, Fig 4E (relative protein levels) considers the ratio between pERK/ERK or is every single protein compared with GAPDH, and then the ratio always pERK(GAPDH)/ERK(GAPDH) made? It is not clear what is expressed in panel E. The same is true for panel G, even if, in that case, it is referred to as % of ctr. However, even in this case, it is not clear how the values in the graphs are evaluated and how the values were calculated.
- Why in Figure 6 in panels A and D the observations are made at different times for the various conditions studied? Perhaps it would be appropriate to make the data presented homogeneous by putting the exact times for all. If significant differences are appreciated at different times, explain better, with additional panels or in the text, or explain why other times are clearly shown.
- In the legend of fig 6, black arrows are indicated, but only white arrows are seen in the figure. Please clarify.
- In panel H of Figure 7, is the GAPDH the same for all the indicated probes? Have the various proteins been evaluated in the same gel? Clarify this point.
- In Figure 8, there is the description of 16 and 39 min, but these times are not present. Clarify and insert them in the figure.
- In Figure 10, there are two drawings of dashed lines with a terminal block and an arrow with a solid line, but there is a solid line with a terminal block in the legend. Which one is valid? Verify well.

Reviewer #3 (Remarks to the Author):

In the manuscript with the title "Protocadherin 15 suppresses oligodendrocyte progenitor cell proliferation and motility via distinct signalling pathways", Yilan Zhen and co-authors describe their findings regarding the role of the transmembrane protein Protocadherin 15 (Pcdh15) in cultured postnatal mouse oligodendrocyte progenitor cells (OPCs). The major highlights of this study are: (1) Pcdh15 inhibits proliferation of OPCs; (2) Pcdh15 promotes contact-mediated cell repulsion and process motility of OPCs; (3) The contact-mediated OPC repulsion and veiling kinetics is mediated by the inhibiting effect of Pcdh15 on the Arp2/3 complex activity.

I think, this is an interesting study, and the message is timely because little is known so far regarding the role of Pcdh15 for the development, behavior, and function of OPCFs. I have few questions and suggestions which the authors may want to consider:

(1) Why are the in situ hybridizations shown in Figure 1 performed in the spinal cord while the rest of the study is carried out using cortical OPCs? I think, it would be more relevant to show in situ hybridizations and/or immunohistochemistry in the mouse cortex in Figure 1. An additional minor issue here is that panels (A)-(C) of the Figure 1 should be brighter, and the brain/ spinal cord sub-regions (e.g. white matter, grey matter, etc) should be appropriately labelled on the panels. It should also be indicated from which regions the shown insets are taken.

(2) In Figure 1F, the pattern of Pcdh15 labelling is exactly the same as the pattern of the PDGFR alpha labelling. Is it expected? Are the authors sure that there is no bleeding through from green channel into the red channel, and vice versa, in their images? Please, show immunohistochemistry for only green and for only red channel (i.e. without double labelling) and verify that the second channel is black. See also my comment #4 below.

(3) If proliferation is increased and motility is decreased upon knockdown of Pcdh15, I am wondering whether in these conditions, more OPCs are located close together after the cell division being unable to migrate away from each other. In other words, does Pcdh15 regulate the speed/chance of migration of recently divided cells away from each other? I think this could be evaluated, for instance, by counting the EdU+ cells which appear as singles, pairs, triples (if any), etc. It would be interesting to present those data in this manuscript.

(4) I have not found a description of the pixel density analysis, neither in the Materials & Methods nor in the Figure Legends. I think it is necessary to include the details of this analysis, also indicating how the confocal images were acquired for this analysis. The background fluorescence in 2H seems to be higher than in 2D and also in 2I-K. Were all those images acquired with the same laser intensity, detector gain, pinhole, etc? Please, include all these details.

(5) I have not found a description of how the EdU-positive cells were defined and counted. The images in Figure 3 show that some cells are brightly stained with EdU and are clearly EdU-positive, while others seem to be weakly labelled and may have only a part of the cell/nucleus labelled. Were those two categories, both counted as EdU-positive?

(6) The authors show that filopodial contact time was increased in Pcdh15 knockdown cultures, and this is mediated by Arp2/3-dependent accumulation of F-actin in filopodia. I am wondering whether Pcdh15 and Arp2/3 are actually expressed at the tips of the OPC filopodia, and whether the observed effects are all completely local, and perhaps represent a part of the purely mechanical sensitivity. Or whether these proteins are rather receivers of the other signals from the other cellular compartments. I think, it would be interesting and important to perform at least immunocytochemistry for Pcdh15 and Arp2/3 specifically in OPCs filopodia in order to shed a bit more light on this issue. For example, the authors could fix the cultures after performing the time-lapse analysis and carry out immunocytochemistry.

Reply to Zhen et al. Reviews

We thank the reviewers for their positive evaluation of our manuscript. We have included new data and respond to each question below.

Reviewer #1

• In Fig2 B,C the reduction in Pcdh15 protein level is very significant and has reduced almost 80-90 % . However, the shRNAs (A,B,C) used independently do not show that much reduction in the Pcdh15 levels in terms of pixel density. Authors may explain or justify the observations.

Yes, treatment of primary mouse OPCs with Pcdh15 shRNA (combination of A, B and C) achieves a high level of knockdown, as shown by Western blot in Fig. 2B and C. When equivalent cultures were fixed and stained to detect Pcdh15 by ICC, the quantification indicated a smaller reduction (Fig. 2D-G), closer to 40%. On page 5 of the results, we indicated that measuring pixel intensity from ICC was not as reliable as measuring protein expression by Western blot but was carried out to confirm knockdown using a smaller number of cells – this is an important consideration when working with primary mouse OPC cultures.

The reviewer particularly asks about the experiment that follows, in which we treat OPC cultures with the separate Pcdh15-shRNA components (A, B and C) in isolation (Fig. 2H-L). When comparing ICC quantification – the separate treatments achieved a similar level of knockdown as the combined Pcdh15-shRNA (compare Fig. 2G with 2L). There is no discrepancy between Pcdh15 shRNA treatment and treatment with the different shRNA components to explain or discuss, but we have edited the results text to indicate the relative reductions in fluorescence on page 5 of the results to ensure this is clear.

By quantifying the integrated pixel density for Pcdh15 immunofluorescence, we determined that OPCs in the *Pcdh15* shRNA-treated cultures had ~36% less fluorescence than control cultures (Fig. 2G; mean ± SEM; n=3 independent cultures analysed per group and ≥ 136 cells analysed per culture; one- way ANOVA, p=0.0005). Furthermore, treating OPC cultures with control shRNA or one of the three components of the commercial *Pcdh15* shRNA in isolation (referred to as A-C), produced a similar reduction in Pcdh15 fluorescence (Fig. 2H-L; ~33%, ~32% and ~30% less fluorescence than control OPCs, respectively). This immunocytochemical approach clearly underestimates the level of knockdown but was used to confirm knockdown in all subsequent functional experiments.

While the relative reductions were similar, pixel density measures were ~5 a.u. higher across the board for the experiment quantified in Fig. 2L than the experiment quantified in Fig. 2G. This is because we maintain imaging parameters within each experiment (all associated cultures) but set them independently for each experiment. This is now described in more detail on page 24 of the methods.

To quantify Pcdh15 expression in OPCs, 40x images were collected from 6 locations spanning each coverslip. The confocal imaging parameters were fixed for each experiment (equivalent for all biological replicas). A control shRNA coverslip was used to set the laser power, exposure time and gain to ensure clear visualisation of the immunocytochemical signal. Those settings were then applied to collect images from all coverslips related to that experiment. Imaging parameters used to collect all images used for quantification reported in Fig. 2G were the same but were not equivalent

to those used to collect images for Fig. 2L. Each image was opened in ImageJ and converted to an 8-bit image. The custom shape tool was used to trace the full outline of each OPC (soma and processes) and the integrated density measured for this region of interest.

• In Fig3, the Edu+/GFP+ cells are not much different in the control and treated groups. Authors do mention that there were 3 independent experiments, however, there is no mention of how many different areas on the slides were scored for counting the Edu+/GFP+ cells to show the robustness of the observation.

There are twice as many EdU+ GFP+ cells in the Pcdh15 shRNA-treated cultures compared to the control cultures. To make this point more clearly, we include split channel images to allow better visualisation of the EdU labelling in Fig. 3 and direct arrows to each EdU labelled cell.

On page 24 of the methods, we indicate that quantification was performed using 20x confocal images collected from 9 locations across each coverslip.

To quantify the proportion (%) of OPCs that express EdU, 20x compressed confocal images were collected from 9 locations spanning each coverslip. EdU is incorporated into the DNA of cells as they transition through S-phase of the cell cycle, however, EdU labelling can result in a mixture of strongly and faintly labelled cells. OPCs will be faintly labelled if, for example, they are already part way through S-phase when EdU labelling commences or have not completed S-phase when EdU labelling concludes. Conversely, EdU labelling can be very strong for OPCs that divide more than once during the labelling period. Each confocal image was opened in ImageJ and the GFP⁺ OPCs manually evaluated and classified as EdU⁺, if any EdU-labelling was detected in the nucleus, or GFP-negative, and these numbers were used to calculate the proportion of EdU-labelled OPCs ($\text{EdU}^+ \text{GFP}^+ / \text{GFP}^+ \times 100$).

On page 5/6 of the results we also include the number of cells evaluated per coverslip (>600 per coverslip).

We found that approximately twice as many GFP⁺ OPCs had divided and incorporated EdU in Pcdh15 knockdown than equivalently treated control cultures (**Fig. 3G**; mean \pm SEM, n=3 independent cultures and ≥ 603 OPCs quantified per coverslip; unpaired *t*-test, p=0.048). A similar increase in OPC proliferation was observed when OPC cultures were treated with any of the 3 *Pcdh15* shRNAs (A-C) in isolation (**Fig. 3H-P**; mean \pm SEM, n=3 independent cultures and ≥ 791 OPCs quantified per coverslip).

• In Fig5F, how one can confirm that the processes shown are the new ones or the old ones. What is the methodology to follow the new process formation versus the existing ones so as to comment whether there is significant reduction in the number of new processes formed.

We agree this was not conveyed well. The images provided for 5F and 6E were single time-point images. We now provide additional data. We have quantified the average number of processes per OPC, which was measured from still images collected from a single time point (Fig. 5E, F), and include additional image series

from our timelapse videos (Fig. 5G and Fig. S2) to better depict new process generation over time.

• In Fig 7D, total and phosphorylated Akt levels do not show any changes in either control or treated groups, however, the authors claimed that Pcdh15 has significant role in cell proliferation.

The reviewer is correct. Our data indicate that Pcdh15 knockdown promotes OPC proliferation by increasing ERK1/2 phosphorylation, not Akt phosphorylation.

ERK1/2 phosphorylation promotes OPC proliferation *in vitro* and *in vivo*, being a key downstream regulator of OPC proliferation induced by PDGFR α , FGF2, BDNF, IGF-1 and NT-3. Indeed, PDGFR α and TrkB promote OPC proliferation without enhancing Akt phosphorylation (Baron et al., 2000; Van't Veer et al., 2009).

The literature strongly argues that Akt/mTOR signalling fine tunes OPC differentiation and myelination rather than proliferation (recently reviewed by Adams et al., 2021). But we now include an additional sentence, considering this point, on page 14 in the discussion.

As Pcdh15 did not alter Akt phosphorylation (Fig. 7), Pcdh15 may be less effective at suppressing OPC proliferation driven by growth factors that simultaneously promote ERK1/2 and Akt phosphorylation, as has been reported for IGF-1³⁶.

• The authors have not discussed about the potential role of Pcdh15 in regulating OPC marker proteins like PDGFR α or PDGFR β levels and their contribution in cell behaviour (proliferation/migration), which will be an important aspect as far as OPC behaviour biology is concerned.

PDGFR β is not highly expressed by OPCs, therefore we have not examined the capacity for Pcdh15 to regulate PDGFR β expression.

PDGFR α is highly expressed by OPCs and PDGF-AA is in the culture medium. Activation of PDGFR α regulates OPC survival, proliferation, and migration. We now include a new figure (Fig. S1). We used ImageJ to convert images of PDGFR α labelling from control and Pcdh15 shRNA cultures to gray scale, selected the cell soma as the region of interest and measured the mean gray value (sum of gray values for all pixels in the selection / total number of pixels). This was not significantly changed following Pcdh15 knockdown.

We also include an additional sentence raising the possibility that Pcdh15 influences receptor expression in the discussion.

It is possible that Pcdh15 achieves this outcome by modulating the expression of mitogenic receptors that activate the MAPK signalling pathway, such as PDGFR α , or that it signals more directly to suppress the MAPK signalling pathway to offsets the effect of mitogenic growth-factors within the environment.

Reviewer #2:

• *How have the choice of concentrations of the various pharmacological agents been evaluated? Have you, by chance, considered any toxic effects on your oligodendrocyte cultures? Cell images do not always show cells in "health" after different treatments.*

The key signs of poor health for cultured OPCs include a failure to proliferate, complete process withdrawal (with no associated process extension) and detachment – and this happens very quickly. As proliferation and process extension are key readouts for our study, these experiments could not be carried out on suboptimal OPC cultures or unhealthy cells. We now include timelapse images showing the OPC cell bodies and new process generation over time (Figs.5, 10 and S2), and these low magnification images also serve to confirm that the OPCs were in good health after ≥ 12 hours of drug exposure.

The final concentration of U0126 in the culture medium was 10 μM . We selected this concentration based on previous studies reporting that it blocks ERK1/2 phosphorylation in primary OPC cultures (Fyffe-Maricich et al., 2011; Tripathi et al., 2017). We now refer to these papers on page 6 of the manuscript.

we treated control and *Pcdh15* knockdown cultures with DMSO (diluent) or the highly selective MEK1/2 inhibitor, U0126^{38,39}, from the time of lentiviral transfection. At a concentration of 10 μM , U0126 prevents MEK1/2 from phosphorylating ERK in primary OPC cultures^{40,41}.

The final concentration of CK666 in the culture medium was 50 μM . We selected this concentration based on a previous study that applied a dilution series of CK666 to primary OPCs under differentiating culture conditions. We selected 50 μM as it was shown to block Arp2/3 and reduce F-actin (phalloidin) labelling without producing signs of toxicity (Zuchero et al., 2015). We now refer to this paper on page 11 of the manuscript.

The final concentration of ML141 in the culture medium was 10 μM . We could not find evidence that this drug had been used to treat primary OPC cultures but found that this dose had been used to effectively block Cdc42 in other cultured cell types, including neurons (Mortal et al., *Frontiers in Neuroscience*, 2017), and refer to this on page 11.

we treated control and *Pcdh15* knockdown cultures with an Arp2/3 inhibitor, CK666 [50 μM ⁶³], or an inhibitor of the upstream Cdc42 GTPase, ML141 [10 μM ⁶⁴], at the time of lentiviral transfection (**Fig. 9A**). At these concentrations, CK666 and ML141 reduce F-actin nucleation in differentiating OPC and neuron cultures, respectively^{44,65}.

• *In the legend of Fig 1 corresponding to panel D, what does "Expression data from 15" mean? Furthermore, the bar indicated in the legend for panels A-C is not present in the figure.*

The graph in panel D was generated from publicly available RNA sequencing data published by Zhang et al. (reference 15). We have extended this sentence in the legend to make this clear.

Graph generated using previously published and publicly available RNA sequencing data from Zhang et al. (2014)¹⁵.

The scale bars for A-C had moved behind the image panels and were not visible. We have now corrected this.

• In Fig 4A, the images of the bands of 5 samples (3 + 2) are shown, but then the densitometries are shown on 3 + 3. For clarity, perhaps it would be appropriate also to show the third sample for Pcdh15 shRNA.

There is no additional lane to show on this example Western blot. We quantified n=2 experiments from the Western blot depicted. The third experiment (control and Pcdh15 shRNA treated cultures) was quantified on a separate Western blot. In the gel photo shown, a positive control shRNA lysate was included in the first lane. The lane was not quantified and has now been cropped from the Figure to avoid confusion.

• The legend of the exact figure is described in a somewhat confusing way. Could you please clarify the description of what is shown in the figure? For example, Fig 4E (relative protein levels) considers the ratio between pERK/ERK or is every single protein compared with GAPDH, and then the ratio always pERK(GAPDH)/ERK(GAPDH) made? It is not clear what is expressed in panel E. The same is true for panel G, even if, in that case, it is referred to as % of ctr. However, even in this case, it is not clear how the values in the graphs are evaluated and how the values were calculated.

Having re-read the legend we understand why the reviewer was confused. We now provide more information in the figure legend for Figure 4. The data were quantified as indicated on the graph axes pERK / ERK to determine the relative proportion of ERK that was phosphorylated.

Figure legend changes:

(E) Western blot protein band integrated pixel density was quantified for p-ERK1/2 and ERK1/2 and used to calculate the proportion of ERK1 that was phosphorylated (p-ERK1/ERK1) and the proportion of ERK2 that was phosphorylated (p-ERK2/ERK2) for OPCs transfected with control- or Pcdh15-shRNA [mean ± SEM, n=3 independent cultures; p-ERK1/ERK1 and p-ERK2/ERK2 data analysed separately by unpaired t-test (note // break to x-axis), *p<0.05, **p<0.01]. (F) Western blot gel image showing ERK1/2, p-ERK1/2, β-actin, and GAPDH expression by OPCs transfected with control- or Pcdh15-shRNA and treated with DMSO or U0126 (10 μM). (G) Quantification of the proportion of ERK1 that was phosphorylated (integrated pixel density for p-ERK1 / integrated pixel density for ERK1) and the proportion of ERK2 that was phosphorylated (integrated pixel density p-ERK2 / integrated pixel density for ERK2) in OPCs transfected with control- or Pcdh15-shRNA and treated with DMSO or U0126. For each culture, these data were normalised to the control shRNA group i.e. relative protein level / control shRNA relative protein level x 100

We also include additional data quantifying ERK expression relative to GAPDH in the results text, to demonstrate that total ERK levels are not affected by Pcdh15 knockdown – only ERK phosphorylation.

we determined that *Pcdh15* knockdown did not affect total ERK1 or ERK2 protein levels, relative to GAPDH expression [mean ERK1/GAPDH \pm SEM for control-shRNA (1.61 ± 0.21) and *Pcdh15*-shRNA (1.76 ± 0.05) treated cultures, $n=3$ independent cultures, unpaired t test, $p = 0.55$; mean ERK2/GAPDH \pm SEM for control shRNA (1.73 ± 0.52) and *Pcdh15* shRNA (1.73 ± 0.33) treated cultures, $n=3$ independent cultures, unpaired t test, $p = 0.99$]. However, *Pcdh15* knockdown was associated with a significant increase in the proportion of ERK1/2 that was phosphorylated (e.g. p-ERK1 / total ERK1 and p-ERK2 / total ERK2; **Fig. 4E**).

To determine whether the increased proliferation detected in *Pcdh15* knockdown OPC cultures was reliant on ERK1/2 phosphorylation, we treated control and *Pcdh15* knockdown cultures with DMSO (diluent) or the highly selective MEK1/2 inhibitor, U0126^{38,39}, from the time of lentiviral transfection. At a concentration of 10 μ M, U0126 prevents MEK1/2 from phosphorylating ERK in primary OPC cultures^{40,41}. After 12 h, protein lysates were generated from the OPC cultures, and a western blot analysis revealed that U0126 treatment had no effect on total ERK1 or ERK2 expression, relative to GAPDH expression [mean ERK1/GAPDH \pm SEM from $n=3$ independent cultures: control-shRNA (0.54 ± 0.11) vs *Pcdh15*-shRNA (0.63 ± 0.08) vs control-shRNA + U0126 (0.62 ± 0.10) vs *Pcdh15*-shRNA + U0126 (0.60 ± 0.06); two- way ANOVA with Bonferroni multiple comparisons test for ERK1/GAPDH: shRNA $F(1,8) = 0.13$, $p = 0.73$; drug treatment $F(1,8) = 0.06$, $p=0.82$; interaction $F(1,8) = 0.33$, $p = 0.58$. Mean ERK2/GAPDH \pm SEM from $n=3$ independent cultures: control-shRNA (0.85 ± 0.18) vs *Pcdh15*-shRNA (0.92 ± 0.07) vs control-shRNA + U0126 (0.95 ± 0.11) vs *Pcdh15*-shRNA + U0126 (0.94 ± 0.06); two- way ANOVA with Bonferroni multiple comparisons test for ERK2/ GAPDH: shRNA $F(1,8) = 0.08$, $p = 0.79$; drug treatment $F(1,8) = 0.30$, $p = 0.60$; interaction $F(1,8) = 0.11$, $p = 0.72$]. However, U0126 treatment markedly reduced ERK1 and ERK2 phosphorylation in control and *Pcdh15* shRNA treated OPCs (**Fig. 4F, G**).

• *Why in Figure 6 in panels A and D the observations are made at different times for the various conditions studied? Perhaps it would be appropriate to make the data presented homogeneous by putting the exact times for all. If significant differences are appreciated at different times, explain better, with additional panels or in the text, or explain why other times are clearly shown.*

Each image series is an excerpt from the corresponding timelapse video, which covers the 2-hour imaging period. Each image series was selected to depict a particular change in morphology relevant to the filopodia, process or cell being imaged. Each image within the series is time-stamped from the first – which we denote as $t=0$. As the timing of events is one of the things that is most altered by *Pcdh15* knockdown, we elected to select stills based on their ability to show a change in the morphology rather than taking images from predetermined timepoints. For these studies it serves our purpose to show that *Pcdh15* knockout veils are extruded for longer, remaining at times well beyond the time that control OPC veils were retracted. We also realised that set timepoints could give a false impression, as a process could be depicted with an extruded veil at one time point and an extruded veil at a later time point – but the retraction that occurred in the middle would be missed. For these reasons, we have not replaced our image series with those from more homogeneous time-frames, but have elected to provide a more detailed explanation of what we quantified in the methods.

Quantification was performed by manually scoring events as present or absent across every video frame (1 min apart) to calculate event duration.

We also ensure that it is clear in the figure legends that the depicted series are excerpts from the larger time-lapse video, and the time stamps do not reflect a duration of filming, but are relative to the first frame shown in the series of stills.

Figure 5: Pcdh15 promotes OPC motility and filopodial repulsion in vitro.

12h after OPCs were transfected with control or *Pcdh15*-shRNA, they were imaged once a minute for 2 h and the resulting time-lapse videos were used to quantify different aspects of OPC motility. (A) Image series show a filopodial contact (black arrowhead) between adjacent OPCs in a control and *Pcdh15* knockdown culture. All images are time-stamped relative to the first image in the depicted series (designated as 0 min). (B) The mean duration (min) that a filopodial contact was sustained once formed between adjacent OPCs transfected in control and *Pcdh15* knockdown cultures (mean \pm SEM, n= 3 independent cultures, \geq 66 filopodial contacts measured per condition; unpaired t test, *p = 0.034). (C) Excerpts from a time-lapse image series showing a control OPC process as it extrudes and retracts a veil and a *Pcdh15* knockdown OPC as it extrudes and maintains its veil. The images are time-stamped relative to the first image in the depicted series (designated as 0 min). Black arrowheads indicate elaborated veils. (D) Quantification of the mean veiling time for OPC processes in control- and *Pcdh15*-shRNA treated cultures (time taken to complete an extrusion and retraction) (mean \pm SEM, n=3 independent cultures, \geq 146 veils analysed per condition; unpaired t-test, **p = 0.004). (E) Images of control- and *Pcdh15*-shRNA treated OPCs, 12 h after transfection. (F) Quantification of the mean number of processes supported by control and *Pcdh15* shRNA-treated OPCs (mean \pm SEM, n=3 independent cultures and \geq 81 OPCs analysed per condition; unpaired t test, ***p = 0.0004). (G) Short image series extracted from the time-lapse videos showing control and *Pcdh15* knockdown OPCs extending a new process. All images are time-stamped relative to the first image in the depicted series (designated as 0 min) and the white asterisks denote the new processes. (H) Quantification of the mean number of new processes elaborated by control- and *Pcdh15*-shRNA transfected OPCs each hour (mean \pm SEM, n=3 independent cultures and \geq 35 OPCs analysed per condition; unpaired t test, *p = 0.048). (I) The migration trajectories for the soma of 52 control- or *Pcdh15*-shRNA treated OPCs. (J) Mean migration speed for the soma of control- and *Pcdh15*-shRNA treated OPCs (mean \pm SEM, n=52 OPCs per group, unpaired t tests with Welch's correction, ****p < 0.0001). (K) Mean distance that each OPC soma moved (path length) in control- and *Pcdh15*-shRNA treated cultures (mean \pm SEM, n=52 OPCs per group, unpaired t tests with Welch's correction, ****p < 0.0001). (L) Mean Euclidean distance (shortest distance between the start and end points) that each OPC soma moved in control- or *Pcdh15*-shRNA treated cultures (mean \pm SEM, unpaired t tests with Welch's correction, ***p < 0.0002). Scale bars represent 5 μ m (A, C), 8 μ m (G) or 20 μ m (E).

Figure 6: Pcdh15-mediated inhibition of ERK does not influence filopodial repulsion or veiling kinetics

(A) Image series extracted from time-lapse videos showing filopodial contacts (arrows) formed between adjacent OPCs in control- and *Pcdh15*-shRNA cultures treated with DMSO or U0126 (10 μ M). All images are time-stamped relative to the first image in the depicted series (designated as 0 min). (B) The mean duration (min) that filopodial contacts were sustained once they form between adjacent OPCs

• In the legend of fig 6, black arrows are indicated, but only white arrows are seen in the figure. Please clarify.

We deleted the word “black” from the figure legend.

• In panel H of Figure 7, is the GAPDH the same for all the indicated probes? Have the various proteins been evaluated in the same gel? Clarify this point.

I thank the reviewer for pointing out this issue. We introduced this error when condensing multiple figures into one. The proteins were quantified from 3 separate gels and we now block each antibody with its corresponding GAPDH bands.

• In Figure 8, there is the description of 16 and 39 min, but these times are not present. Clarify and insert them in the figure.

These values were taken from the graph in Fig. 8Q (now Fig. 9Q) and correspond to the mean veiling time measured for OPC processes in each condition, across 3 independent cultures. We now make that clearer in the results text.

By reviewing timelapse videos of OPC processes over time, as they extrude and retract lamellipodia-like veils, we were able to quantify the time taken for individual veils to be initiated, extruded and fully retracted (Fig. 9Q, n=3 independent cultures and ≥ 122 processes measured per treatment condition). We found that the mean veiling time for vehicle-treated control OPCs was ~16 min and this increased to ~39 min in *Pcdh15* knockdown cultures (Fig. 9Q; image series showing example veil extrusions and retractions are in Fig. 9R). When *Pcdh15* knockdown cultures were treated with ML141, veiling time was significantly shortened (~12 min; Fig. 9Q), indicating that *Pcdh15* is a negative regulator of *cdc42*-Arp2/3 signalling and is necessary for the rapid motility of actin rich OPC structures.

As the reviewer indicates, the corresponding image series in Fig. 9R does not show 16 minutes of stills from a control movie or 39 min of stills from a *Pcdh15* knockdown movie. Instead we show example stills from a 17 min time-course of a control process, in which the veil has not yet been elaborated at t=0 and has been fully retracted by 17 min. The white arrows indicate the frames in which the veil is extruded. We also show example stills of a *Pcdh15* knockdown process, demonstrating that the veil is still extruded at 21 min. The control shRNA + ML141 and *Pcdh15* shRNA + ML141 image series show good examples of veil extrusion and retraction (a complete veiling event) in a timeframe consistent with the graphical data.

We now provide additional information in each figure legend.

Figure 9: *Pcdh15* suppresses Arp2/3 activity to promote filopodial repulsion and process veiling by OPCs.

(A) A schematic proposing a mechanism by which *Pcdh15* could impact actin polymerisation in OPCs and depicting the point of action in the pathway of the drugs ML141 and CK666. (B-M) Confocal images of OPC processes in control- and *Pcdh15*-shRNA cultures, treated with diluent (DMSO), CK666 (50 μ M) or ML141 (10 μ M), and immunolabelled to detect microtubules (α - tubulin, green) and F-actin (Phalloidin, red). (N) Phalloidin intensity (integrated pixel density) in the veils of OPCs from control- and *Pcdh15*-shRNA cultures treated with DMSO, CK666 or ML141 [mean \pm SEM, n= 3 independent cultures and ≥ 50 veils analysed per treatment condition; two-way ANOVA: shRNA treatment F (1,12) = 5.22, p = 0.041; drug treatment F (2,12) = 86.54, p<0.0001; interaction F (2,12) = 13.14, p = 0.0009 with Bonferroni multiple comparisons, *p<0.01, ****p<0.0001]. (O) Image series, extracted from time-lapse videos, depicting filopodia from adjacent OPCs making

contact in control- and *Pcdh15*-shRNA cultures treated with DMSO or without ML141. Arrows indicate contacting filopodia. All images are time-stamped relative to the first image in the depicted series (designated as 0 min) and images are selected to highlight the formation and maintenance of filopodial contacts. (P) The mean duration that OPC filopodia remain in contact (min) in control- and *Pcdh15*-shRNA cultures treated with DMSO or ML141 [mean \pm SEM, n= 3 independent cultures, \geq 61 filopodial contacts analysed per treatment condition; two-way ANOVA: shRNA treatment F (1,8) = 63.09, $p < 0.0001$; drug treatment F (1,8) = 135.0, $p < 0.0001$; interaction F (1,8) = 40.95, $p < 0.0001$ with Bonferroni multiple comparisons, $*p < 0.05$, $****p < 0.0001$]. (Q) Quantification of the mean time taken for an OPC process to extrude and retract a veil in control- and *Pcdh15*-shRNA cultures treated with DMSO or ML141 [mean \pm SEM, n= 3 independent cultures, \geq 122 veils quantified per treatment condition; two-way ANOVA: shRNA treatment F (1,8) = 47.69, $p = 0.0001$; drug treatment F (1,8) = 84.96, $p < 0.0001$; interaction F (1,8) = 21.94, $p = 0.0016$ with Bonferroni multiple comparisons, $*p < 0.05$, $***p < 0.001$, $****p < 0.0001$]. (R) Image series selected from time-lapse videos of control- and *Pcdh15*-shRNA OPCs treated with DMSO or ML141, showing OPC processes as they extrude and retract veils. All images are time-stamped relative to the first image in the depicted series (designated as 0 min) and images are selected to highlight veil extrusion and retraction. Arrows indicate elaborated veils. Scale bars represent 2 μ m (B-M) or 5 μ m (O, R).

• In Figure 10, there are two drawings of dashed lines with a terminal block and an arrow with a solid line, but there is a solid line with a terminal block in the legend. Which one is valid? Verify well.

We have corrected this in the figure legend

Reviewer #3:

(1) Why are the in situ hybridizations shown in Figure 1 performed in the spinal cord while the rest of the study is carried out using cortical OPCs? I think, it would be more relevant to show in situ hybridizations and/or immunohistochemistry in the mouse cortex in Figure 1. An additional minor issue here is that panels (A)-(C) of the Figure 1 should be brighter, and the brain/ spinal cord sub-regions (e.g. white matter, grey matter, etc) should be appropriately labelled on the panels. It should also be indicated from which regions the shown insets are taken.

We now demarcate the border of the white and grey matter regions of the spinal cord for panels A-C and indicate where the insets were taken from. While we appreciate the reviewer's other suggestions, we have not replaced the spinal cord in situs with images from the brain. We had good reason to carry out our earliest analysis in the spinal cord and not the brain. Firstly, the spinal cord pMN domain produces OPCs in embryonic development that persist postnatally. This is not true for the earliest OPCs produced in the embryonic brain. Secondly, the timing of OPC migration in the spinal cord allowed us to select a timepoint for analysis at which OPCs are the major cell type present in the white matter. This is more challenging in the developing brain. As our spinal cord in situ data are consistent with the RNA sequencing data for cells isolated from the early postnatal cortex and our own cell-specific RNA sequencing from the adult mouse brain, we do not feel that in situ

hybridisation performed in the developing mouse brain will provide significantly more insight into *Pcdh15* expression by OPCs.

(2) In Figure 1F, the pattern of Pcdh15 labelling is exactly the same as the pattern of the PDGFR alpha labelling. Is it expected? Are the authors sure that there is no bleeding through from green channel into the red channel, and vice versa, in their images? Please, show immunohistochemistry for only green and for only red channel (i.e. without double labelling) and verify that the second channel is black. See also my comment #4 below.

We had noticed the similarity between the location of PDGFR α labelling and *Pcdh15* labelling in OPCs. However, they do not exactly colocalise and the staining pattern remained, even when OPCs were not co-labelled to detect PDGFR α . However, the reviewer may be more reassured by our inclusion of a new figure (Fig. 8), which shows single confocal scans through primary mouse OPCs labelled with 2 distinct *Pcdh15* antibodies. The first detects all forms of *Pcdh15*, the other detects CD3 isoforms of *Pcdh15* - the isoforms expressed by OPCs. Even in these single confocal scans, *Pcdh15* (green) shows polarised expression to one side of the soma and expression in the processes. This staining pattern is similar to that of PDGFR α , but these cells were not labelled to detect PDGFR α .

Modified results text on page 10:

As actin dynamics can be regulated by local protein activity, we next confirmed that *Pcdh15* and members of the Arp2/3 complex were expressed throughout OPC processes. We performed immunocytochemistry to detect *Pcdh15* (anti-pan *Pcdh15*, green) and Arp2 or Arp3 (red) in primary OPC cultures and collected high magnification, single plane confocal images for analysis (Fig. 8A-H). We found that *Pcdh15* expression was highly concentrated to one side of the cell soma (Fig. 8A-C, E-G), but was also identified as immunofluorescent puncta along the length of each process (Fig. 8D, 8H). Puncta corresponding to Arp2 (Fig. 8A-D) and Arp3 (Fig. 8E-H) were also clearly visible in the OPC soma and along each process. This pattern of labelling was further confirmed by performing immunocytochemistry using antibodies to detect Arp2 or Arp3 and an antibody specific to CD3 isoforms of *Pcdh15*²⁹ (Fig. 8I-P), the major *Pcdh15* isoforms expressed by OPCs (see Fig. 1E). As *Pcdh15* and Arp2/3 complex proteins are expressed at the OPC soma and within OPC processes, it is feasible that they are distinct components of a local signalling pathway that regulates actin dynamics in OPCs.

(3) If proliferation is increased and motility is decreased upon knockdown of Pcdh15, I am wondering whether in these conditions, more OPCs are located close together after the cell division being unable to migrate away from each other. In other words, does Pcdh15 regulate the speed/chance of migration of recently divided cells away from each other? I think this could be evaluated, for instance, by counting the EdU+ cells which appear as singles, pairs, triples (if any), etc. It would be interesting to present those data in this manuscript.

We now include additional data relating to the quantification of EdU+ nuclei in pairs etc and the distance between EdU+ nuclei in Fig. 3 (new panels 3Q and 3R) as requested.

New results text:

A more detailed analysis of the EdU⁺ GFP⁺ nuclei revealed that they more frequently appeared as doublets in Pcdh15 knockdown cultures i.e pairs of EdU⁺ GFP⁺ nuclei in direct contact (**Fig. 3Q**), and the mean Euclidean distance between EdU⁺ GFP⁺ cells was reduced (**Fig. 3R**). These data are consistent with Pcdh15 acting as a negative regulator of OPC proliferation *in vitro*.

New methods text:

To calculate the average distance between EdU⁺ nuclei, each image converted to a 16-bit image, inverted, a threshold set to ensure that all EdU⁺ nuclei were detected, and the Euclidean distance (EDM nuclei) measured in ImageJ.

It is difficult to say whether the increase in doublets is simply the result of the increased proliferation or also a slowing of OPC basal motility affecting the movement of the daughter cells away from the mother cells. We also quantified the basal motility of the OPC soma and found that basal motility was slightly reduced in Pcdh15 knockdown cultures (new data in Fig. 5I-K).

New results text:

While these data suggest that Pcdh15 knockdown OPCs have an abnormal morphology that is supported by altered process kinetics, this phenotype may relate to a broader change in OPC basal motility. Under proliferative culture condition, the soma of each OPC does not move a great distance. However, by mapping the migration trajectories of individual OPCs (following the movement of the cell soma over time; **Fig. 5I**) we determined that speed of soma movement (**Fig. 5J**), the total distance each cell moves (**Fig. 5K**) and the Euclidean distance i.e. total distance from soma start to end position (**Fig. 5J**) were all reduced in Pcdh15 knockdown cultures.

New methods text:

The basal motility of cultured OPCs was quantified from time-lapse video files containing images collected every min for 2 h. The video files were opened in ImageJ (National Institutes of Health, USA) and tracing of the cell soma performed using the Manual Tracking plugin (Fabrice Cordelières, Institut Curie, Orsay, France). OPC motility is minimal under these culture conditions. However, OPCs were selected by researchers who were blind to the culture conditions and assigned for motility tracing if they were within the central third of the imaging field at the onset of image collection (reduced the likelihood of the cell moving outside of the imaging field) and were near the edge or outside of an OPC cluster. Each track was analysed using the Chemotaxis and Migration Tool 2.0 (Ibidi GmbH, Munich, Germany). The software was also used to generate trajectory plots, calculate speed, and quantify total distance moved and the Euclidean distance (distance from start to end point as a direct line).

(4) I have not found a description of the pixel density analysis, neither in the Materials & Methods nor in the Figure Legends. I think it is necessary to include the details of this analysis, also indicating how the confocal images were acquired for this analysis. The background fluorescence in 2H seems to be higher than in 2D and also in 2I-K. Were all those images acquired with the same laser intensity, detector

gain, pinhole, etc? Please, include all these details.

Yes, the background fluorescence was different for the two experiments presented in Figure 2, as the microscopy settings were equivalent within each experiment (i.e. all replica cultures), but not between experiments. Our analysis was performed on the raw image files. We set the laser power, gain and exposure time for each experiment – but we did not set the pinhole, as images were collected on a spinning disk confocal microscope.

We now describe this in more detail on page 24 of the manuscript and include more information about the associated image analysis.

To quantify Pcdh15 expression in OPCs, 40x images were collected from 6 locations spanning each coverslip. The confocal imaging parameters were fixed for each experiment (equivalent for all biological replicas). A control shRNA coverslip was used to set the laser power, exposure time and gain to ensure clear visualisation of the immunocytochemical signal. Those settings were then applied to collect images from all coverslips related to that experiment. Imaging parameters used to collect all images used for quantification reported in Fig. 2G were the same but were not equivalent to those used to collect images for Fig. 2L. Each image was opened in ImageJ and converted to an 8-bit image. The custom shape tool was used to trace the full outline of each OPC (soma and processes) and the integrated density measured for this region of interest.

(5) I have not found a description of how the EdU-positive cells were defined and counted. The images in Figure 3 show that some cells are brightly stained with EdU and are clearly EdU-positive, while others seem to be weakly labelled and may have only a part of the cell/nucleus labelled. Were those two categories, both counted as EdU-positive?

EdU labelling results in faintly and brightly labelled cells. We add EdU to the culture medium for 12 hours. It will be incorporated into the DNA of all cells during the DNA synthesis phase of the cell cycle. Cells that were already part way through S-phase at the time the EdU was added, will incorporate EdU into their DNA, but will be labelled faintly (as it was only incorporated into some of the DNA). Similarly, cells that enter S-phase just before the cells are fixed and have not completed S-phase will only be faintly labelled. By contrast cells that divide once or more while EdU is present will have their DNA strongly labelled.

We want to quantify the total number of cells that were in S-phase (divided) within our labelling period. All EdU+ cells are counted irrespective of the level of EdU labelling.

We now include more information about how EdU+ labelling is defined and quantified on page 24 of the methods section of the paper.

To quantify the proportion (%) of OPCs that express EdU, 20x compressed confocal images were collected from 9 locations spanning each coverslip. EdU is incorporated into the DNA of cells as they transition through S-phase of the cell cycle, however, EdU labelling can result in a mixture of strongly and faintly labelled cells. OPCs will be faintly labelled if, for example, they are already part way

through S-phase when EdU labelling commences or have not completed S-phase when EdU labelling concludes. Conversely, EdU labelling can be very strong for OPCs that divide more than once during the labelling period. Each confocal image was opened in ImageJ and the GFP⁺ OPCs manually evaluated and classified as EdU⁺, if any EdU-labelling was detected in the nucleus, or GFP-negative, and these numbers were used to calculate the proportion of EdU-labelled OPCs ($\text{EdU}^+ \text{GFP}^+ / \text{GFP}^+ \times 100$). To calculate the average distance between EdU⁺ nuclei, each image converted to a 16-bit image, inverted, a threshold set to ensure that all EdU⁺ nuclei were detected, and the Euclidean distance (EDM nuclei) measured in ImageJ.

(6) The authors show that filopodial contact time was increased in Pcdh15 knockdown cultures, and this is mediated by Arp2/3-dependent accumulation of F-actin in filopodia. I am wondering whether Pcdh15 and Arp2/3 are actually expressed at the tips of the OPC filopodia, and whether the observed effects are all completely local, and perhaps represent a part of the purely mechanical sensitivity. Or whether these proteins are rather receivers of the other signals from the other cellular compartments. I think, it would be interesting and important to perform at least immunocytochemistry for Pcdh15 and Arp2/3 specifically in OPCs filopodia in order to shed a bit more light on this issue. For example, the authors could fix the cultures after performing the time-lapse analysis and carry out immunocytochemistry.

We now include a new Figure (Fig. 8) in which we have stained OPC primary cultures to detect Pcdh15 and Arp2 or 3. We include high magnification images of the OPC processes showing that all 3 proteins are expressed throughout OPC processes.

REVIEWERS' COMMENTS:

Reviewer #1 (Remarks to the Author):

All the points/queries raised earlier are not only answered but the detailing has been done very diligently. The corrections have been incorporated in the edited manuscript version and all needed justifications wherever expected has been provided.

Overall, the manuscript has been significantly enhanced.

The data and findings of the present study will be very useful for our fundamental understanding of Oligodendrocyte progenitor biology.

Reviewer #2 (Remarks to the Author):

I thank the authors who modified the article, following the reviewers' advice, changing the text, figures and legends. Now the paper is ready to be published.

Reviewer #3 (Remarks to the Author):

The authors have addressed my comments and have answered my questions. The revised version of the manuscript, in which the authors have taken into consideration the suggestions of all the reviewers, appears more complete and stronger. I think, the results of this study will be interesting for researchers studying development of the oligodendrocyte lineage cells and myelination, as well as for a broader research community interested to understand the mechanisms of cell proliferation during normal and pathological conditions.